# Non-invasive electromechanical assessment during atrial fibrillation identifies underlying atrial myopathy alterations with early prognostic value

Electromechanical characterization during atrial fibrillation (AF) remains a significant gap in the understanding of AF-related atrial myopathy. This study reports mechanistic insights into the electromechanical remodeling process associated with AF progression and further demonstrates its prognostic value in the clinic. In pigs, sequential electromechanical assessment during AF progression shows a progressive decrease in mechanical activity and early dissociation from its electrical counterpart. Atrial tissue samples from animals with AF reveal an abnormal increase in cardiomyocytes death and alterations in calcium handling proteins. High-throughput quantitative proteomics and immunoblotting analyses at different stages of AF progression identify downregulation of contractile proteins and progressive increase in atrial fibrosis. Moreover, advanced optical mapping techniques, applied to whole heart preparations during AF, demonstrate that AF-related remodeling decreases the frequency threshold for dissociation between transmembrane voltage signals and intracellular calcium transients compared to healthy controls. Single cell simulations of human atrial cardiomyocytes also confirm the experimental results. In patients, non-invasive assessment of the atrial electromechanical relationship further demonstrate that atrial electromechanical dissociation is an early prognostic indicator for acute and long-term rhythm control.

Atrial fibrillation (AF) is a complex arrhythmia that often starts with short-duration events and progresses to persistent and long-lasting episodes[1]. AF itself induces progressive functional and structural changes in the atrial myocardium that facilitate long-term perpetuation of the arrhythmia[2,3]. These changes include alterations to ion channels that shorten action potential duration and progressively lead to faster atrial activation rates until a state of complete electrical remodeling is reached[4]. In parallel, the atria progressively dilate, further decreasing the likelihood of successful rhythm control[5]. Remodeling also affects the mechanical properties of atrial myocytes[6],

resulting in contractile dysfunction associated with AF recurrences and early cardioembolic events after cardioversion[7,8]. Atrial myopathy may also precede AF onset, establishing a favorable substrate for recurrences and long-term arrhythmia maintenance[9].

Electrical remodeling can be evaluated non-invasively using surface ECG tracings after appropriate signal processing to subtract ventricular complexes and specifically assess atrial activation rates during AF[10]. Conversely, assessing mechanical activity during AF is challenging, and most reported data have evaluated atrial deformation properties during sinus rhythm by tissue Doppler imaging

✉ e-mail: david.filgueiras@cnic.es

(TDI)[7], transmitral pulsed Doppler[11], or more recently speckle tracking[12]. Some reports have used tissue velocity imaging to assess mechanical activation rates (MAR) at specific atrial spots during AF, showing a prognostic association with AF termination and recurrences[13,14]. Moreover, results from a small series suggested that tissue velocity–imaging-derived MAR may correlate with local intracavitary electrogram activations in AF[15]. However, as AF-related changes progress, calcium channel remodeling[2,6,16], ryanodine receptor refractoriness[17], and the decrease in mechanical activity might result in electrical activations occurring faster than their mechanical counterparts, a phenomenon that could be defined as electromechanical dissociation (EMD). Therefore, more advanced tools to assess the electromechanical relationship are needed to address current gaps in knowledge and to define the pathological process involved in each AF patient[1], especially at early disease stages before overt signs of atrial myopathy.

We propose that EMD is an early indicator of atrial remodeling progression during AF that influences acute cardioversion rates and sinus rhythm maintenance. Here, we studied electromechanical remodeling progression in vivo in a long-term pig model of AF using a non-invasive approach based on simultaneous acquisition of surface electrocardiography recordings and TDI signals. In isolated whole heart preparations, we investigated the underlying mechanisms on advanced optical mapping studies of simultaneous transmembrane voltage and intracellular free calcium during AF. In atrial tissue samples, we obtained further mechanistic insights from molecular biology analyses, immunohistochemistry and histopathology studies, which were complemented with blood biomarkers of atrial damage and single cell simulations to understand the biological process associated with AF-related EMD. Finally, we validated the prognostic value of non-invasive assessment of the atrial electromechanical relationship in a clinical series of symptomatic patients at early stages of AF-related remodeling.

## Results

### Electromechanical assessment shows highly synchronized activity in the healthy atria

A group of 16 pigs with healthy atria underwent atrial electromechanical assessment during programmed atrial stimulation and short-lasting AF episodes. All pigs underwent AV node ablation before atrial pacing or AF to prevent very rapid AV node conduction[18]. Back-up ventricular pacing was set at 40 beats per minute (bpm). TDI sequences were obtained from transesophageal echocardiography (TEE) views (Fig. 1a) focused on the posterior left atrial wall ($N = 16$) at the highest possible frame rate (>200 Hz). All local TDI signals from every pixel within the segmented atrial wall (Fig. 1b) were averaged to generate higher quality signals over a 6-s acquisition segment. Lead II ECG data were acquired simultaneously. Programmed atrial stimulation at 300, 250 and 200 ms cycle length showed that TDI-derived waves followed the corresponding pacing frequencies (Fig. 1c), without differences between electrical activation rates (EAR) and the simultaneous counterpart TDI-derived MAR (Fig. 1d). The highest peak in the power spectral density (i.e., dominant frequency; DF) of both the lead II-derived atrial ECG and the atrial TDI signals was used to obtain a robust and rapid calculation of the average EAR and MAR, respectively (Fig. 1c). Short-lasting AF episodes after 10-s burst atrial pacing at 10 Hz (median AF episodes duration: 23.5 s [11.5, 56.0 s]) showed that EAR also followed MAR (Fig. 1e). Slightly faster EAR (<1.35 Hz) than the TDI-derived mechanical counterparts were only documented in 4 of 36 movies (Fig. 1f).

In 7 of 16 pigs, simultaneous Lead II ECG tracings and TDI signals were obtained from the posterior left atrium and the left atrial free wall (Fig. 1g). In the healthy atria, this regional analysis showed no signs of EMD in any of these 2 left atrial regions (Fig. 1h).

### Electromechanical assessment shows overt dissociation after long-lasting AF

A group of 11 pigs were implanted with a dual chamber pacemaker with atrial and ventricular leads inserted into the right atrial appendage and right ventricular apex, respectively. After 10 days of recovery, all pigs underwent AV node ablation before attempting high-rate atrial pacing (HRAP). The pacemakers were programmed to induce AF using an automatic algorithm until the AF became self-sustaining and HRAP was no longer necessary (Fig. 2a). After a median of 12.7 months [9.3, 14.4] of self-sustained AF and 27.0 months [24.4, 35.7] of protocol initiation, pigs with AF showed overt signs of atrial remodeling with larger atrial areas (indexed to weight) than sham-operated controls (22.9 cm² [21.0, 25.3] vs 15.7 cm² [14.9, 16.7], $p < 0.001$, Supplementary Fig. 1). All pigs with persistent AF underwent an invasive electrophysiological study to assess local electromechanical activity under 3D electroanatomical guidance (Fig. 2a, b). The analysis of simultaneous TDI signals and spatially colocalized local unipolar electrograms demonstrated overt EMD between EAR and TDI-derived MAR (Fig. 2b–d). EMD was also documented on regional analyses at the posterior left atrium and the left atrial free wall (Fig. 2e). Similar results were obtained using more global assessment of EAR based on DF values of lead II-derived atrial ECG (Fig. 2f). In fact, during the electroanatomical mapping procedures 70.9 ± 11.8% of the atrial endocardial surface showed local DF values within the 5th–95th percentiles of lead II-derived atrial EAR (Suppl. Fig. 2). This further supports the use of lead II tracings as a valid reference for actual EAR on most of atrial regions. Further analysis using a mixed model based on Generalized Estimating Equations (GEE) demonstrated that both DF values from lead II-derived atrial ECG ($p < 0.001$) and persistent AF ($p < 0.001$) exerted a statistically significant effect on EMD. Moreover, the effect of atrial EARs on EMD was significantly different between pigs with and without AF-related remodeling (interaction $p < 0.001$, Fig. 2f).

### Time-course of electromechanical remodeling during AF progression in pigs

A group of 13 pigs underwent sequential monitoring of remodeling progression at 3-week intervals after the initiation of the AF induction protocol (Fig. 3a). A further 6 animals were used as sham-operated controls. As per study protocol, the starting point for electromechanical assessment (i.e., t0) was defined at the time of the first acquisition of simultaneous TDI and lead II signals during an AF episode lasting ≥6 s. The latter took 3-to-6 weeks after the initiation of the AF induction protocol.

In AF animals, transthoracic echocardiography (TTE) studies showed progressively larger left atrial areas compared to sham-operated controls (Fig. 3b). In TEE studies, electromechanical assessment from t0 onwards showed a steady increase in DF values of lead II-derived atrial ECG tracings, whereas simultaneous TDI-derived DF values progressively decreased and dissociated from the electrical activity (Fig. 3c–e). Overall, EMD reached 1.9 ± 1.7 Hz and statistical significance ($p = 0.014$) at 9 weeks after the first recorded episode ≥6 s (i.e., t0). Similar to previous reports in sheep and patients with AF[2,4], electrical remodeling showed a steady increase in DF values until electrical remodeling reached a plateau phase without further increase in EARs (21 to 27 weeks after t0. Fig. 3f). Interestingly, 12 of 13 pigs (92%) showed EMD before the follow-up reached 27 weeks after t0 (Fig. 3g). In fact, EMD was documented at the time of 89.1 ± 19.9% AF burden. The remaining animal did not show EMD during the entire follow-up period (48 weeks after t0) and its AF episodes did not reach persistent AF criteria (≥7 days in self-sustained AF). Overall, in vivo data indicate that EMD is an early indicator of AF remodeling progression and follows animal-specific patterns.

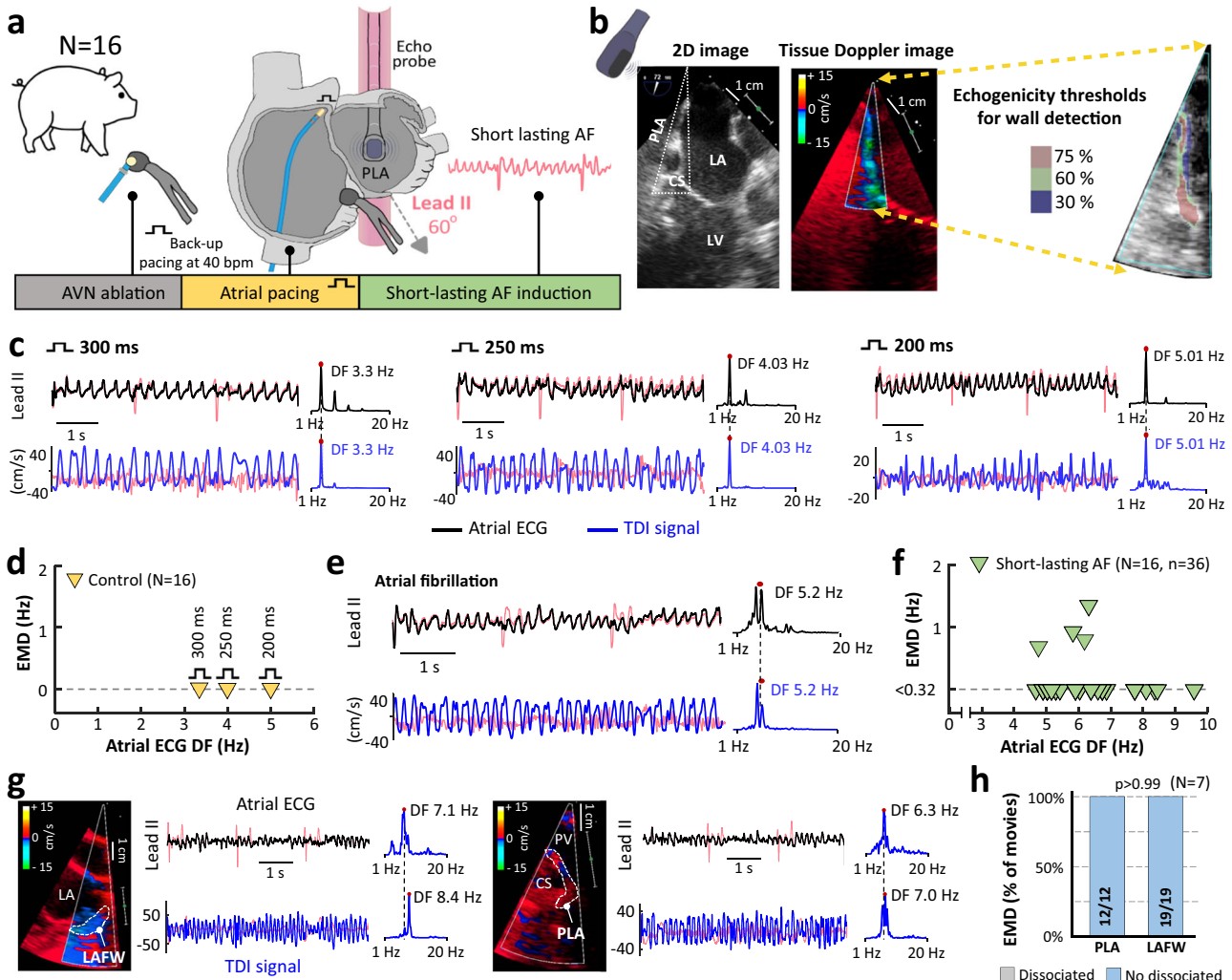

**Fig. 1 | Electromechanical assessment in the healthy atria of pigs. a** Schematic experimental flow-chart. **b** Two-dimensional and tissue Doppler imaging (TDI) snapshots (during acquisition and segmentation) of the posterior left atrial wall (PLA) during transesophageal echocardiography imaging in a pig. **c** Representative simultaneous TDI- and lead II-derived atrial signals during programmed atrial stimulation at 300, 250 and 200 ms cycle length. Their respective power spectral density and dominant frequency (DF) peaks are also shown. **d** Quantification of electromechanical dissociation (EMD = DF of atrial ECG signal − DF of atrial TDI signal) in all animals undergoing programmed atrial stimulation. **e** Representative TDI- and lead II-derived atrial signals during short-lasting atrial fibrillation (AF) after burst pacing. **f** Quantification of EMD during short-lasting AF episodes in control

animals with healthy atria ('*n*' indicates the number of movies for analysis). **g, h** Regional analysis of EMD using TDI signals from the PLA and the left atrial free wall (LAFW). Based on the spectral resolution of atrial ECG and TDI signals, the EMD threshold was set at 0.32 Hz, which ensures proper distinction of the two DF peaks (See Suppl. Material for details). The Fisher's exact test was used to assess differences. Light red tracings (in **c, e, g**) show the original tracings before subtracting QRS-T complexes from lead II and the intrinsic mode functions from TDI signals (See Methods for details). AVN atrio-ventricular node, CS coronary sinus, LA left atrium, LV left ventricle, PV pulmonary vein. Source data are provided as a Source Data file.

## Defective muscle contraction is early established during AF remodeling

A group of 4 pigs underwent sequential right atrial biopsies at 3 time-points during the follow-up: baseline, after 6 weeks of 100% AF burden and at the end of follow-up after >6 months in persistent AF (Fig. 4a). High-throughput quantitative proteomics showed a significant decrease in proteins belonging to Gene Ontology categories associated with muscle contraction (Suppl. Table 1), already at the time-point of 6 weeks of 100% AF burden (Fig. 4b). The proteomics analysis also showed a significant increase in proteins associated with collagen-containing extracellular matrix and structural resistance of the extracellular matrix (Fig. 4b, Suppl. Table 1). Histopathology studies in the atrial biopsies from AF animals showed a significant increase in fibrosis compared to samples from sham-operated controls at equivalent time-points of the follow-up (Fig. 4c, d). Moreover, plasma levels of pro-collagen type III N-peptide (PIIINP) were also higher in AF animals than

in sham-operated controls at equivalent time-points of follow-up (Fig. 4e). Gene expression analysis of lysyl oxidase in atrial biopsies also supported a progressive increase in collagen resistance, which was significantly higher in AF animals than in sham-operated controls at the end of follow-up (Fig. 4f). Defective contractile properties during AF were further confirmed in atrial biopsy samples with immunoblots of the cardiac isoform of myosin-binding protein-C (MYBPC3) (Fig. 4g, h). Conversely, actin alpha cardiac muscle-1 (ACTC1) did not show any significant changes over the follow-up of both AF animals and sham-operated controls (Fig. 4g, h). Altogether, these data support early atrial remodeling changes during AF progression that affect muscle contraction and may contribute to EMD in vivo.

## AF is associated with underlying atrial cardiomyocyte death

Based on the fact that during AF the atria are activated at very rapid frequencies, this should be associated with high metabolic demands,

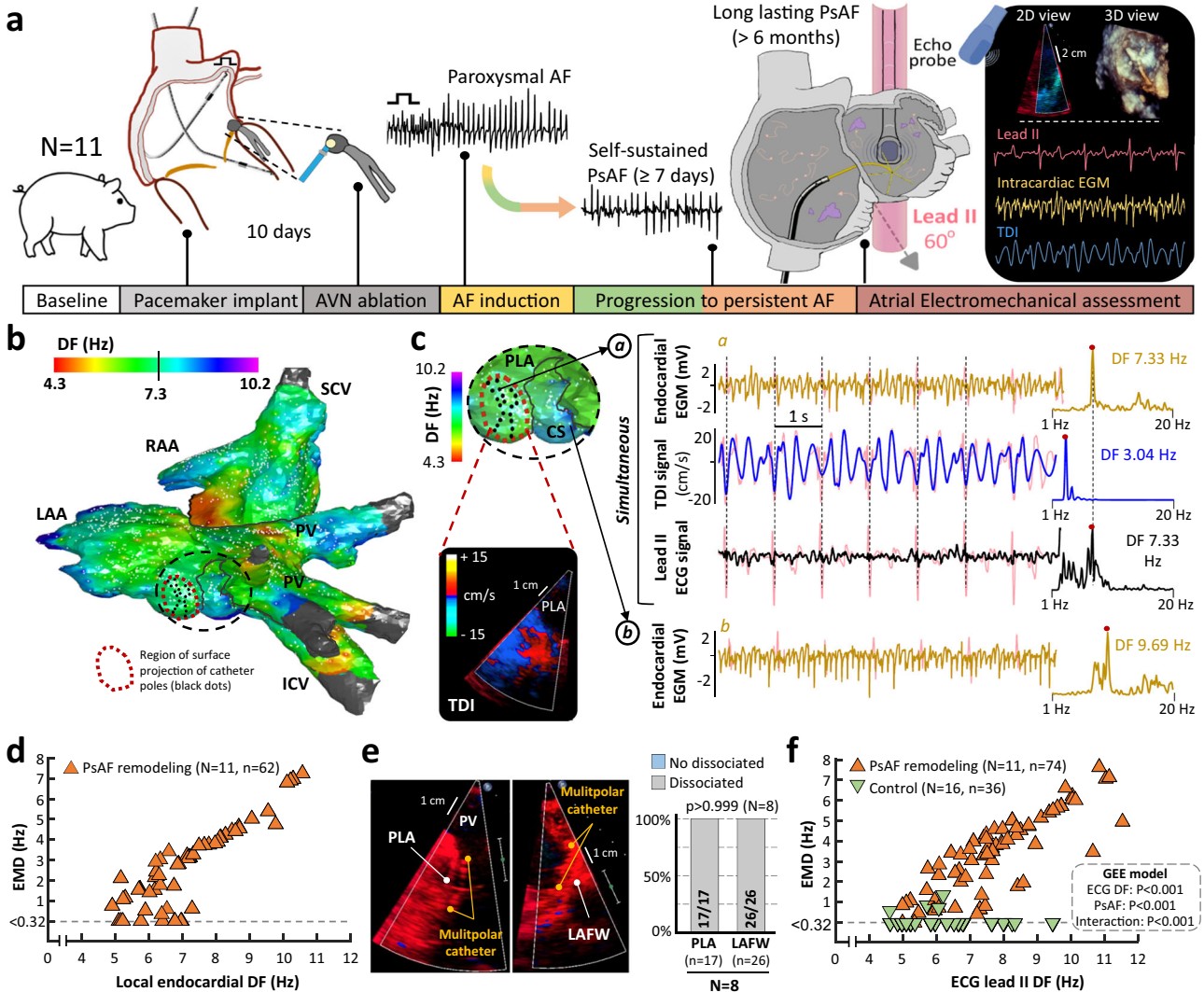

**Fig. 2 | Electromechanical assessment in the atria of pigs with long-lasting atrial fibrillation. a** Schematic experimental flow-chart. **b, c** Sample high-resolution DF map, 2D TDI snapshot and representative tracings from the region with simultaneous acquisition of endocardial electrograms (EGMs) and TDI signals. Simultaneous lead II tracings (in black) are also shown. Light red tracings (in c) show the original tracings before signal processing. **d** Quantification of EMD during invasive electroanatomical mapping. Intracardiac EGMs from a 20-pole catheter were spatially colocalized with the field of view for TDI. Local EMD was calculated as follows: average DF of unipolar EGMs − DF of atrial TDI signal. **e** Regional analysis of EMD using TDI signals from the PLA and the LAFW. The Fisher's exact test was used to assess differences. **f** Quantification of EMD using the DF of the lead II-derived atrial ECG signal (as in Fig. 1). Green triangles indicate data from the control animals (in Fig. 1f). A mixed model based on Generalized Estimating Equations (GEE) was used to test for significant differences and interactions. I/SVC inferior/superior vena cava, LAA/RAA left/right atrial appendage, PsAF persistent AF. DF, EMD, LAFW, TDI, PLA, and PV as in Fig. 1. Source data are provided as a Source Data file.

which could increase cell apoptosis in the atria and further contribute to contractile dysfunction. Interestingly, in AF animals, plasma levels of high-sensitivity TroponinT (hs-TnT) significantly increased during the follow-up, especially during persistent AF stages and compared to sham-operated controls (Fig. 5a). Immunohistochemistry analysis of atrial and ventricular samples from persistent AF animals revealed a significantly higher percentage of Caspase3+ apoptotic cells in the left atrium than in the left ventricle (Fig. 5b). The lack of ventricular damage in AF animals was also confirmed with echocardiography studies during the follow-up, which did not show any significant differences in left ventricular ejection fractions between AF animals and sham-operated controls (Suppl. Fig. 3). Moreover, histopathology analysis in ventricular samples also showed no differences in interstitial fibrosis content between the two groups (Fig. 5c). Further immunohistochemistry and confocal microscopy images of atrial samples revealed a higher number of TUNEL+ cells in AF animals than in sham-operated controls (Fig. 5d). Immunoblots also showed a significant increase in the proapoptotic mediator BAX in atrial samples

from AF animals compared to sham-operated controls. Interestingly, the left atria of both groups also showed higher levels of BAX than their corresponding left ventricular samples (Fig. 5e). Altogether, these data support that AF is associated with low levels of continuous atrial damage and myocyte death that may also contribute to contractile dysfunction.

## AF remodeling decreases the frequency threshold for voltage and calcium dissociation

Simultaneous optical mapping of transmembrane voltage and intracellular free calcium was performed in isolated Langendorff-perfused hearts of animals with persistent AF (10.0 months [5.0, 13.2] of self-sustained AF, $N = 6$) and control animals without atrial remodeling ($N = 7$). Optical recordings were obtained on the left atrial free wall using a 3-mm diameter optical fiber positioned against the atrial surface at several sequential locations. We developed a ratiometric approach using di-4-ANEQ(F)PTEA and Fura-2AM that permits optical mapping in the contracting tissue (Suppl. Fig. 4), without the use of

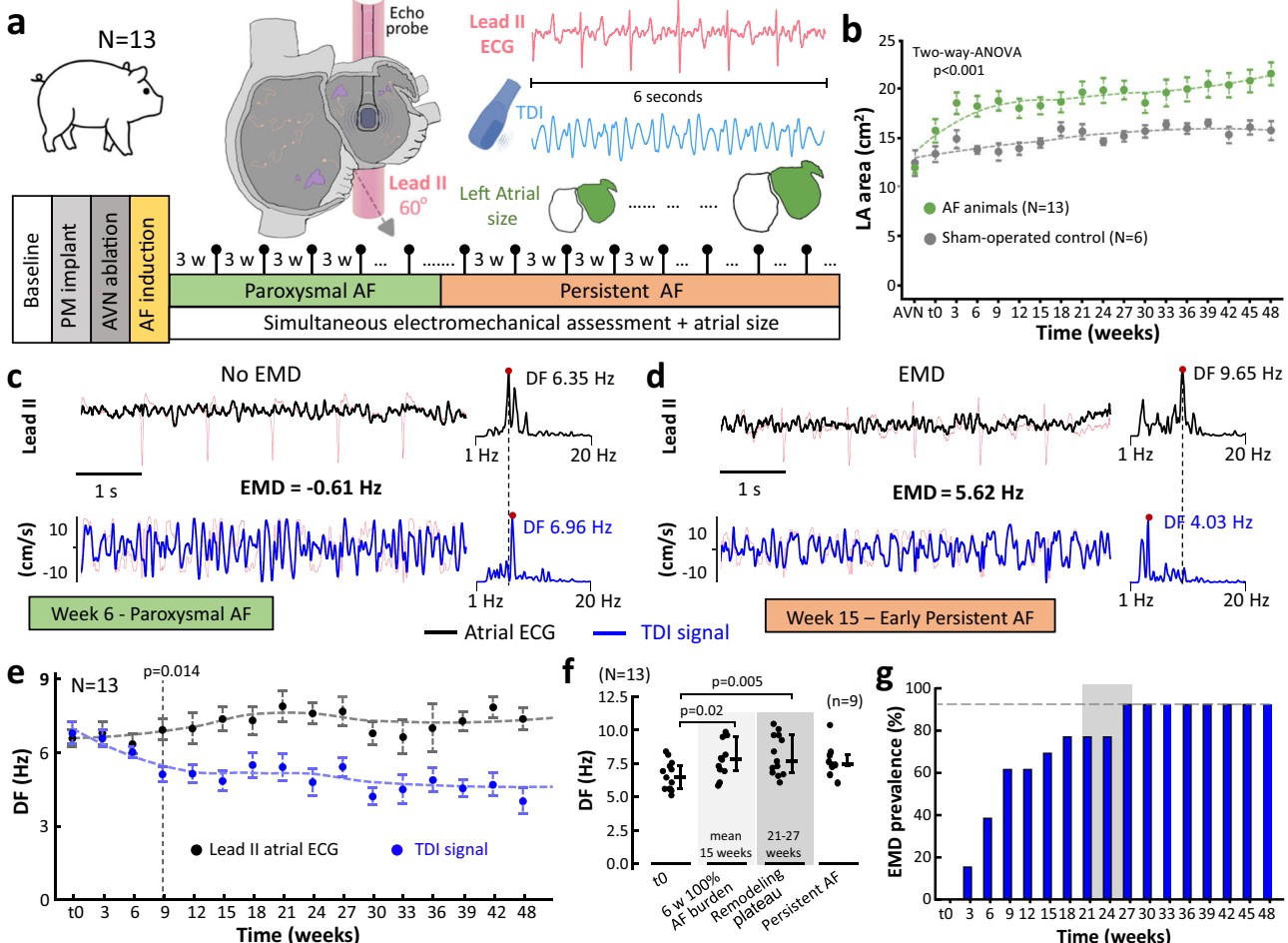

**Fig. 3 | Time-course of atrial electromechanical remodeling during atrial fibrillation progression in pigs. a** Schematic experimental flow-chart.
**b** Comparisons of left atrial (LA) areas between atrial fibrillation (AF) animals and sham-operated controls over the follow-up. **c, d** Sample lead II-derived atrial ECG tracings and their simultaneous TDI-derived signals taken at 6 (**c**) and 15 (**d**) weeks (w) after the first recorded episode lasting ≥6 s (t0). Their respective power spectral density and DF peaks are also shown. Light red tracings show the original signals before processing. **e** Time-course of electromechanical remodeling during AF progression in the pig model. Dashed lines show time-course trends for each parameter. At the beginning of AF history (t0), DF values of TDI signals from atrial

sampling regions were occasionally higher than DF values from lead II, reflecting the more global nature of atrial surface activations on lead II. **f** DF analysis of lead II-derived atrial ECG signals at different time-points during the study protocol. Box-plots show median and interquartile range. **g** 92% of pigs showed EMD by the time electrical remodeling reached a plateau phase between week 21 and 27 (gray vertical bar). Deviation intervals in **b** and **e** represent standard errors of mean values. Two-sided unpaired/paired Student's *t* tests were used to assess differences in (**e**) and (**f**), respectively. Source data are provided as a Source Data file. PM pacemaker. AVN, DF, EMD and TDI as in Fig. 1.

excitation-contraction uncouplers. Since atrial contraction distorts fluorescence signals by the same factor for both excitation wavelengths in excitation ratiometry, taking the ratio of the numerator and denominator signals eliminated motion artifacts (Fig. 6a). In hearts from animals with persistent AF and controls, atrial pacing at 300, 250 and 200 ms cycle length did not show any differences in DF values from transmembrane voltage and calcium transients over 6-s segments (Fig. 6b, c). In hearts from AF animals, the atrial pacing protocol was performed after mapping AF and cardioversion. Sample tracings during pacing at 300 ms are shown in Suppl. Fig. 5.

Optical mapping recordings during AF showed a frequency-dependent dissociation between voltage and calcium signals (Fig. 6d, e). Importantly, at fibrillatory frequencies <6.5 Hz, the hearts from animals with persistent AF showed significantly higher dissociation between voltage and calcium signals than controls (Fig. 6f). Similar to in vivo results using TDI and lead II ECG signals (Fig. 2f), a mixed model based on GEE demonstrated that both DF values from voltage signals and persistent AF exerted a significant effect on voltage and calcium dissociation (Fig. 6e). Moreover, the effect of activation frequency on calcium dissociation was

significantly different between pigs with and without underlying AF-related remodeling (Fig. 6e, interaction *p* < 0.001). In 8 of 13 hearts (2 from persistent AF animals), we also increased the fibrillatory frequency by adding acetylcholine into the perfusate (0.5 μM and 1 μM). At fibrillatory frequencies >8.5 Hz, voltage and calcium dissociation was documented in 100% of the optical mapping tracings for both control and persistent AF hearts (Fig. 6f). In 5 of 7 (71%) control and 4 of 6 (67%) persistent AF hearts, dissociation was preceded by calcium transient alternans (Suppl. Fig. 6a). Tracings that included abrupt changes in activation rates showed that intracellular calcium transients commonly decreased in amplitude or became negligible at high activation frequencies (Suppl. Fig. 6b, c).

Overall, optical mapping data suggest that AF remodeling affects calcium dynamics, which may exacerbate EMD in vivo. In fact, in right atrial samples, immunoblots of key calcium handling proteins showed a significant increase in total phospholamban (PLN) and a decrease in PLB phosphorylation at Thr17 (Thr17-pPLN) in AF animals compared to sham-operated controls. The ratio Thr17-pPLN/Total-PLN also decreased significantly in AF animals (Suppl. Fig. 7a). Both Ryanodine receptor2 (RyR2) channels and RyR2 phosphorylation at Ser2814

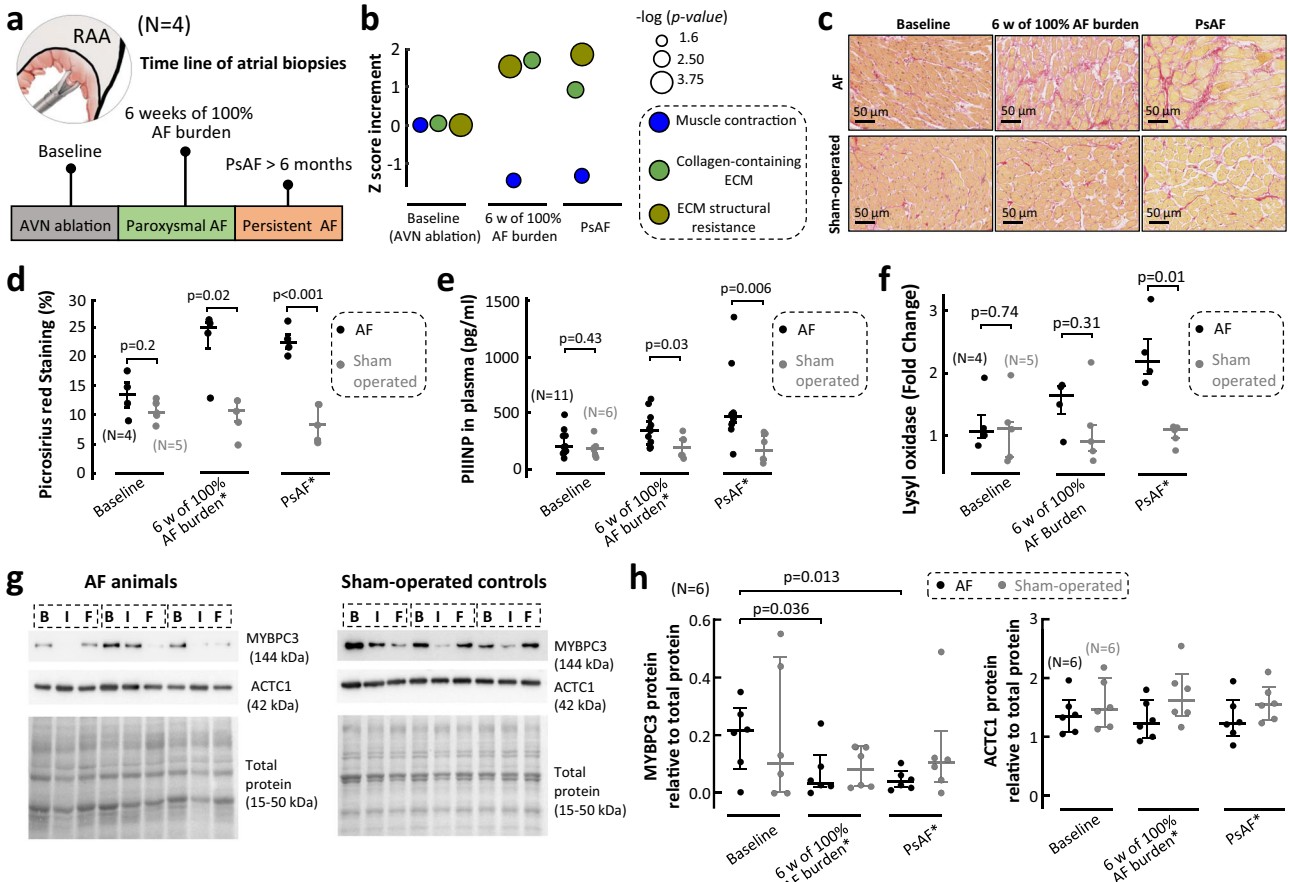

**Fig. 4 | Fibrotic and contractile remodeling during atrial fibrillation progression in pigs. a** Time line of atrial biopsy studies. **b** Atrial fibrillation (AF)-related time-course changes of proteins belonging to muscle contraction, collagen-containing extracellular matrix (ECM) and ECM structural resistance. Color-coded circles represent the average of $Zq$ values of the proteins within each category, normalized to the baseline, that show a statistically significant change over the follow-up. A paired one-way-ANOVA test was used for comparisons. The rest of proteins with statistically significant changes over the follow-up are shown in Suppl Table 1. **c** Representative examples of myocardial atrial biopsies (at 40X) from AF animals and sham-operated controls stained with Picrosirius red at different time-points of follow-up. **d, e, f** Comparisons of fibrosis quantification (**d**), plasma pro-collagen type III N-peptide (PIIINP) (**e**) and gene expression levels of lysyl oxidase (**f**) between AF animals and sham-operated controls. Two-sided unpaired Student's

$t$ tests were used to assess differences. **g** Representative immunoblots of actin alpha cardiac muscle-1 (ACTC1) and the cardiac isoform of myosin-binding protein-C (MYBPC3) in AF animals and sham-operated controls. **h** Quantification and comparisons of ACTC1 and MYBPC3 expressions between AF animals and sham-operated controls. Immunoblot bands were quantified relative to total protein. *indicates equivalent time-points of the follow-up for sham-operated controls. 'B', 'I' and 'F', in (**g**) indicate Baseline, intermediate (i.e., 6 weeks of 100% AF burden or equivalent in controls) and final (i.e., persistent AF > 6 months or equivalent in controls) ($N = 6$). Data were depicted as individual values with median and inter-quartile range. Data were analyzed with repeated measures one-way ANOVA followed by the Dunnett's multiple comparisons test for follow-up comparisons. Source data are provided as a Source Data file. PsAF persistent AF, RAA right atrial appendage.

(Ser2814-pRyR2) showed a significant decrease in AF animals compared to sham-operated controls. However, the ratio Ser2814p-RyR2/Total-RyR2 did not show statistically significant differences between AF animals and controls (Suppl. Fig. 7b). Additional analysis of protein expressions of the sarco-endoplasmic reticulum ATPase2 (SERCA2) and the sodium/calcium exchanger (NCX) did not show differences between groups (Suppl. Fig. 7c, d).

To further understand the role of the underlying remodeling on voltage and calcium dissociation during AF we performed single cell simulations of human atrial action potentials and their corresponding calcium transients. We specifically tested 3 scenarios; one without remodeling (based on the model by Skibsbye et al.[19]), and two with AF-related remodeling, one of which included ion current changes plus additional calcium handling changes associated with the -82% decrease in Thr17-pPLN/Total-PL documented in the experimental data (Suppl. Fig. 8a, b). Simulation outcomes showed significantly higher values of voltage and calcium dissociation in the models with atrial remodeling compared to simulations without remodeling (Suppl. Fig. 8c, d).

Altogether, optical mapping data, the changes in calcium handling proteins and single cell simulations support that voltage and calcium dissociation is not only a frequency dependent phenomenon but it is significantly modulated by underlying remodeling changes.

## Electromechanical dissociation is a predictor of acute rhythm control in patients at early stages of AF remodeling

Similar to experimental data in pigs, atrial electrical remodeling also follows individual-specific patterns in patients[4]. In fact, the analysis of atrial electrical remodeling in patients has suggested that the first 6 months in AF represent a temporal window with special relevance to identify early indicators of remodeling, before more advanced signs of atrial myopathy are established[9].

Here, we conducted a prospective clinical study with patients at early remodeling stages of AF progression (episode duration ≤6 months based on symptoms) and without other relevant comorbidities. The study flow-chart is shown in Fig. 7a (See methods for details). Patients under baseline treatment with antiarrhythmic drugs (including calcium channel blockers and digoxin) other than β-

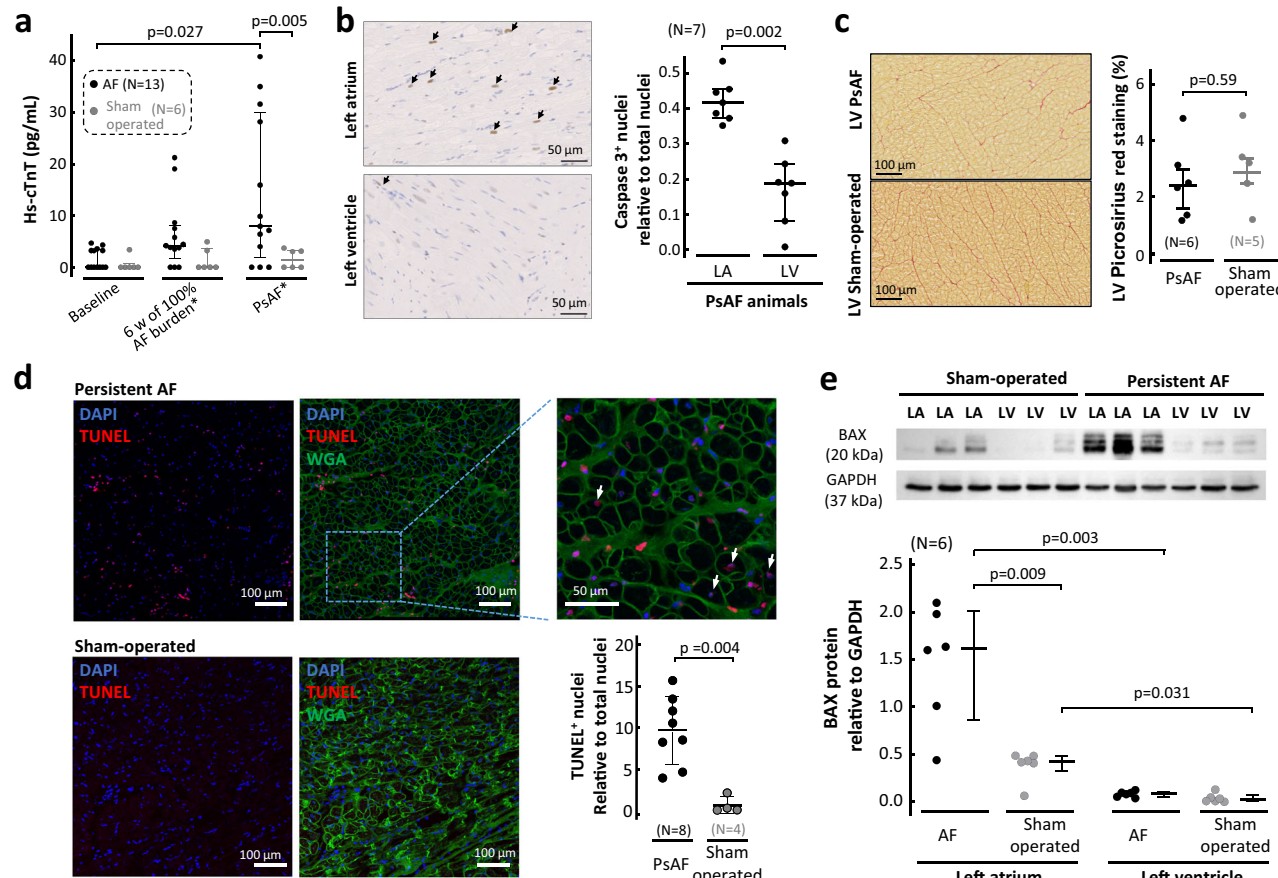

**Fig. 5 | Identification of atrial myocardial damage during atrial fibrillation.**
**a** Quantification and comparisons of high-sensitivity TroponinT (hs-TnT) levels in plasma samples of atrial fibrillation (AF) animals and sham-operated controls over the follow-up. Statistical significance was tested with 2-way ANOVA followed by the Šídák's multiple comparisons test ($p = 0.027$). **b** Representative images and quantification of Caspase3 staining in left atrial (LA) and left ventricular (LV) samples of animals with persistent AF (PsAF). Caspase3+ immunoreactivity was visualized with DAB stain (in brown, indicated with black arrows) and cell nuclei were visualized with hematoxylin (in blue). A two-tailed paired Student's *t* test was used to assess differences. **c** Sample Picrosirius red staining and interstitial fibrosis quantification in LV samples of AF animals and sham-operated controls. A two-tailed unpaired Student's *t* test was used to assess differences. **d** Detection and quantification of apoptosis by TUNEL assay in LA samples from AF animals and sham-operated controls. Samples were counterstained for nuclei (blue, DAPI) and wheat germ agglutinin (green, WGA). White arrows indicate atrial cardiomyocytes with TUNEL+ nuclei. The Mann–Whitney test was used to assess differences. **e** Representative Western blots (upper) showing the expression of BAX against GAPDH in LA and LV samples from AF animals and sham-operated controls. Quantification and comparisons of BAX expression relative to GAPDH. Data are displayed as individual values with median and interquartile range. Mann–Whitney's and Wilcoxon's tests were used to assess statistical significance between groups and anatomical regions, respectively. Source data are provided as a Source Data file.

blockers were excluded. Simultaneous TDI and surface lead II recordings were obtained during TTE and TEE (if indicated) to assess the atrial electromechanical relationship over 6-s segments (as in the animal protocol). The primary outcome was cardioversion to sinus rhythm within a 24-h window after flecainide administration orally.

The study population ($N = 83$) was composed of relatively young patients ($55.8 \pm 9.8$ years old) with low rates of comorbidities (Table 1 and Fig. 7b) and average 3D left atrial volume index values ($32.6 \pm 8.6$ ml/m²) below the dilation threshold ($34$ ml/m²)[20]. In 65% of patients ($n = 54$), the AF episode was the first one detected. The majority (71%) of patients with persistent AF had no previous AF history. Episode duration before attempting pharmacological cardioversion was 18 h [10.0, 30.0 h] in paroxysmal AF patients and 60.0 days [16.0, 121.7 days] in patients with persistent AF. Altogether, these data indicate an AF population at early remodeling stages, which minimized the risk of potential confounding factors affecting atrial remodeling beyond AF itself.

Cardioversion to sinus rhythm was documented in 43 patients of the entire population (51.8%) and in 20 patients with persistent AF (35%) after a median of 150 min [75, 240 min] after flecainide orally. Transient hypotension and conversion to atrial flutter were

documented in 1 and 3 patients, respectively. EMD values from both TTE and TEE studies were significantly different between patients with and without successful pharmacological cardioversion (Table 1 and Fig. 7c). Conversely, the mean amplitude excursion of TDI signals from the left or right atrium did not show any significant differences between groups (Table 1 and Suppl. Fig. 9). Table 1 also shows the results of the univariate analysis of other clinical variables, blood biomarkers, atrial strain parameters and other conventional echocardiography- and ECG-derived parameters.

In the entire AF population, electromechanical assessment during non-invasive TTE demonstrated that a model based on EMD provided higher sensitivity, specificity and area under the curve (79.3%, 79.3% and 0.874, respectively) to predict the primary outcome than AF classification, or multivariate clinical- or ECG-based models (Fig. 7d). The odds ratio for each variable is shown in Fig. 7e. EMD also provided the highest increase in certainty for the primary outcome when added to AF classification (Fig. 7f). Similar to experimental results (Fig. 2f and Fig. 6e), a mixed model based on GEE demonstrated that both DF values from lead II-derived atrial ECG and persistent AF exerted a significant effect on EMD (Fig. 7g). Moreover, the effect of DF values from lead II-derived atrial ECG on

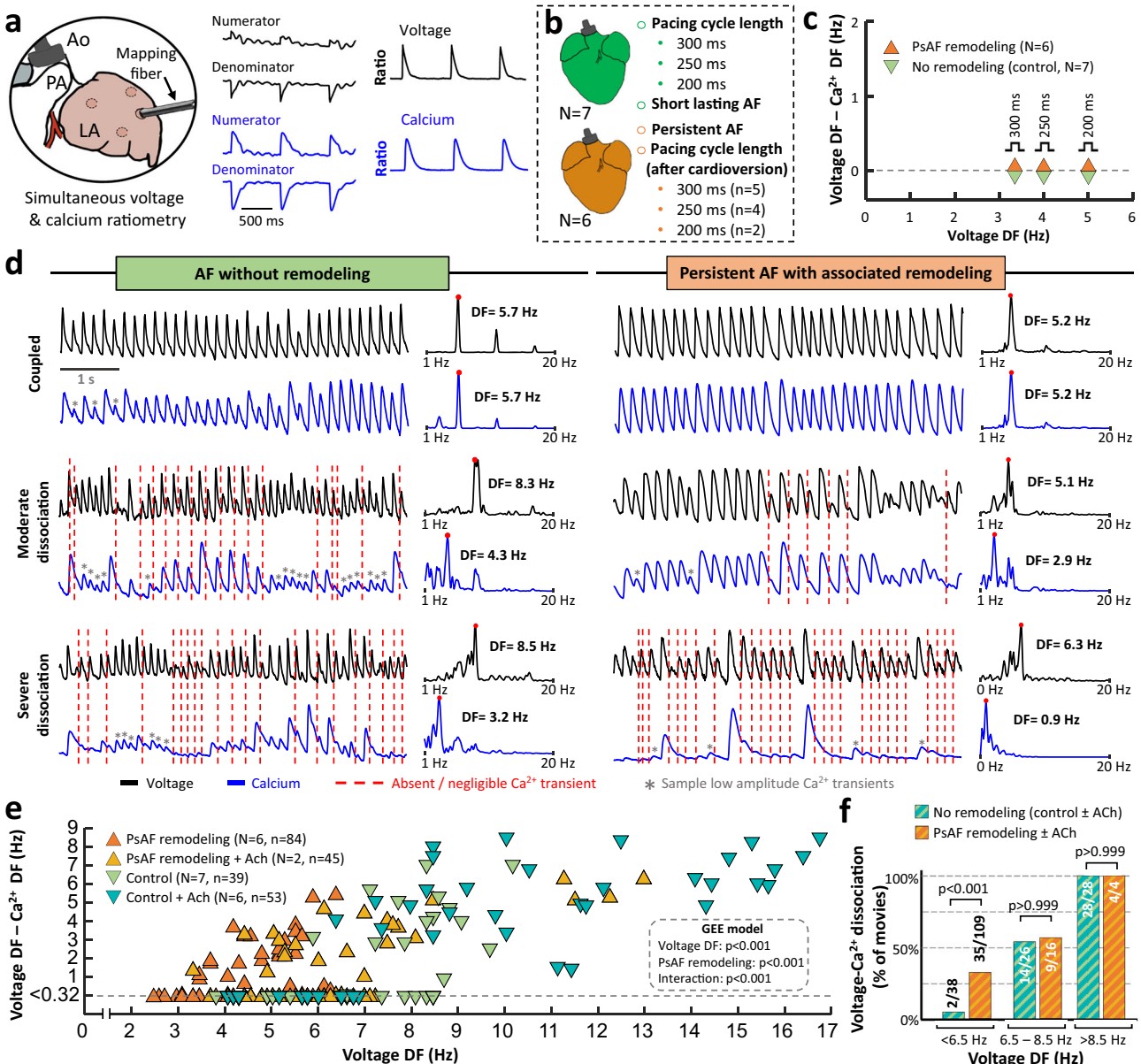

**Fig. 6 | Optical mapping of simultaneous voltage and intracellular calcium in the contracting pig heart. a, b** Schematic experimental setting for simultaneous voltage and calcium ratiometry (**a**) and flow-chart (**b**). Dye loading details are provided in the Suppl. Methods. **c** Quantification of differences between dominant frequency (DF) values of voltage and calcium signals during programmed atrial electrical stimulation. Programmed atrial stimulation in the hearts of animals with persistent atrial fibrillation (PsAF) was performed after voltage and calcium ratiometry during AF and subsequent cardioversion. **d** Sample 6-s voltage and calcium signals during AF in isolated Langendorff-perfused hearts from controls and persistent AF animals. Their respective power spectral density and DF peaks are also shown. **e** Quantification of differences between DF values of voltage and calcium signals during AF. For consistency with in vivo electromechanical assessment, the dissociation threshold for voltage/calcium optical mapping signals was also set at 0.32 Hz. A mixed model based on Generalized Estimating Equations (GEE) was used to test for significant differences and interactions. **f** Prevalence of voltage and calcium dissociation in optical movies of hearts from controls and AF animals at 3 different voltage frequency ranges. The Fisher's exact test was used to assess differences. All optical movies were taken at 500 frames per second. Source data are provided as a Source Data file. ACh acetylcholine, Ao aorta, LA left atrium, PA pulmonary artery.

EMD was significantly different in patients with paroxysmal and persistent AF (Fig. 7g, interaction *p* < 0.001).

## Influence of electromechanical dissociation on secondary clinical outcomes

The study secondary outcomes were pharmacological cardioversion of persistent AF and AF recurrences at 1-year follow-up (Fig. 7a). In the subset of patients with persistent AF, electromechanical assessment during non-invasive TTE demonstrated that a model based on EMD provided a higher sensitivity, specificity and area under the curve (84.6%, 81.5% and 0.875, respectively) to predict pharmacological

cardioversion than a multivariate clinical- or ECG-based model (Fig. 7h). Consistent with the notion that electromechanical assessment provides a more objective and patient-specific characterization of the underlying atrial remodeling state, EMD was also a better predictor of acute pharmacological cardioversion in persistent AF patients than a model based on episode duration (AUC 0.875 vs 0.730, respectively). Sample representative cases with persistent AF and electromechanical assessment using simultaneous lead II-derived atrial ECG and TDI-derived signals are shown in Fig. 8a, b, and Suppl. Movies 1, 2.

After cardioversion, all patients underwent an initial rhythm control strategy including antiarrhythmic drug therapy as a first step

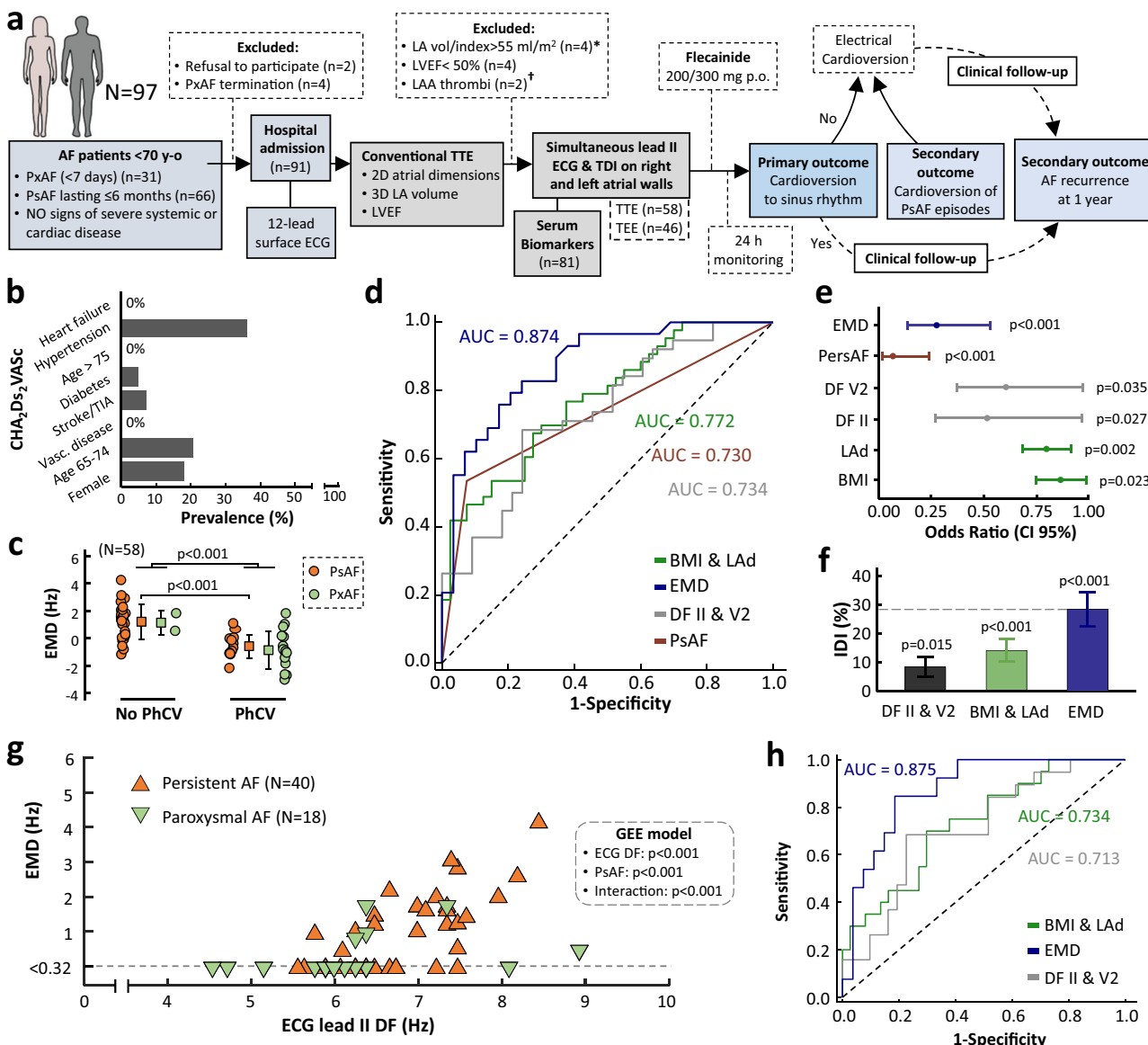

**Fig. 7 | Clinical value of electromechanical assessment in patients at early stages of atrial fibrillation progression. a** Schematic flow-chart and study outcomes. *, † one patient in each of these groups also had left ventricular ejection fraction (LVEF) < 50%. **b** Prevalence of each of the CHA₂Ds₂VASc variables. **c** Comparisons of electromechanical dissociation (EMD) values between patients with successful and failed pharmacological cardioversion (PhCV). The square and whiskers show mean and standard deviation. Two-sided Student's *t* test was used to assess differences. **d** Receiver operating characteristic curves (ROC) of four models for predicting acute PhCV. **e** Odds ratio and Confidence Intervals95% for each variable shown in the selected models (in **d**). Wald's test was used to assess statistical significance. **f** Integrated discrimination improvement (IDI) upon addition to the AF classification: the electrical model (DF leads II and V2, $N = 70$), the clinical model (body mass index [BMI] and left atrial diameter [LAd], $N = 80$), and the EMD ($N = 58$). Bars and whiskers show IDI and standard error. The statistical significance of IDI was assessed with a Z test. **g** Quantification of EMD in patients with paroxysmal (PxAF) and persistent AF (PsAF). A mixed model based on Generalized Estimating Equations (GEE) was used to test for significant differences and interactions. **h** ROC curves of the models in (**d**) to predict acute PhCV in the subset of patients with PsAF. Data normality was assessed with the Shapiro–Wilk test. Source data are provided as a Source Data file. TIA: transient ischemic attack. TEE transesophageal echocardiography, TTE transthoracic echocardiography.

before attempting catheter ablation (Fig. 9a). Importantly, patients with AF recurrences at 3 and 12 months of follow-up showed significantly higher EMD dissociation values than those patients without recurrences (Fig. 9b, c, respectively). Kaplan–Meier curves and log-rank test analysis also showed higher AF recurrences at 1-year follow-up ($p = 0.002$) and 2.5-year follow-up ($p = 0.003$) in patients with EMD values $\geq 0.32$ Hz (Fig. 9d). In contrast, atrial dilation using 3D left atrial volume index or right atrial area criteria (34 ml/m² or 11 cm²/m²)[20,21] did not differentiate between patients with or without AF recurrences during the follow-up (Fig. 9e, f). Similar results were obtained in the subset of persistent AF patients (Fig. 9g–i).

Consistent with a general strategy of attempting rhythm control in study participants, persistent AF patients with higher EMD values also showed a statistically significant association with the subsequent need of $\geq 1$ antiarrhythmic drug and AF ablation during the follow-up (Suppl. Fig. 10a). These association was not statistically significant using right or left atrial dilation criteria (Suppl. Fig. 10b, c).

Altogether, the clinical results further supported the relevance of EMD as an early indicator of atrial remodeling before other changes in conventional echocardiography parameters or blood biomarkers (Table 1) became relevant.

**Table 1 | Baseline characteristics and univariate analysis**

| Study variables on admission | Study population (N = 83) | Pharmacological cardioversion (n = 43) | N Non-Pharmacological cardioversion (n = 40) | p-value |
|---|---|---|---|---|
| Age (years) | 55.8 ± 9.8 | 55.2 ± 11.0 | 56.3 ± 8.3 | 0.986 |
| Sex male/female (n (%)) | 68/15 (81.9/18.1) | 35/8 (81.4/18.6) | 33/7 (82.5/17.5) | 0.896 |
| Clinical history | | | | |
| Body mass index (kg/m$^2$) | 28.4 ± 7.9 | 26.4 ± 3.3 | 30.4 ± 10.4 | 0.001 |
| Lean body mass (Kg) | 57.2 ± 7.9 | 56.0 ± 8.3 | 58.5 ± 7.2 | 0.140 |
| Hypertension (n (%)) | 30 (36.1) | 12 (27.9) | 18 (45.0) | 0.105 |
| Dyslipidemia (n (%)) | 30 (36.1) | 15 (34.9) | 15 (37.5) | 0.804 |
| Diabetes mellitus (n (%)) | 4 (4.8) | 4 (9.3) | 0 (0) | 0.117 |
| Current smoking (n (%)) | 10 (12.1) | 5 (11.6) | 5 (12.5) | 1.000 |
| Previous stroke or TIA (n (%)) | 6 (7.2) | 2 (4.7) | 4 (10.0) | 0.422 |
| Persistent AF (n (%)) | 57 (68.7) | 20 (46.5) | 37 (92.5) | <0.001 |
| CHA$_2$DS$_2$-VASc (≥1 for ♂ / > 1 for ♀) | 39 (47.0) | 18 (41.9) | 21 (52.5) | 0.332 |
| Baseline medication (n (%)) | | | | |
| β-blockers | 58 (69.9) | 29 (67.4) | 29 (72.5) | 0.616 |
| ACEis/ARBs/MRAs | 17 (17.0) | 7 (16.3) | 10 (25.0) | 0.325 |
| Statins | 15 (18.1) | 8 (18.6) | 7 (17.5) | 0.896 |
| Oral anticoagulation | 45 (54.2) | 18 (41.9) | 27 (67.5) | 0.019 |
| Echocardiography parameters | | | | |
| Left ventricular ejection fraction (%) | 60.0 ± 4.8 | 60.4 ± 4.4 | 59.7 ± 5.2 | 0.499 |
| LA diameter; Parasternal long-axis (mm) | 40.2 ± 4.2 | 38.3 ± 4.1 | 42.1 ± 3.4 | <0.001 |
| RA area (cm$^2$/m$^2$) | 9.9 ± 2.0 | 9.4 ± 2.0 | 10.4 ± 1.9 | 0.029 |
| 3D LA volume index (ml/m$^2$) | 32.6 ± 8.6 | 30.6 ± 9.1 | 34.7 ± 7.6 | 0.034 |
| Mean peak flow velocity in LAA (cm/s) | 35.0 ± 13.9 | 38.7 ± 14.3 | 29.9 ± 11.8 | 0.060 |
| LA mean TDI amplitude excursion (cm/s) | 10.8 ± 3.9 | 11.0 ± 3.5 | 10.6 ± 4.3 | 0.699 |
| RA mean TDI amplitude excursion (cm/s) | 11.6 ± 5.1 | 12.5 ± 4.4 | 10.8 ± 5.6 | 0.262 |
| LASr (%) | 13.0 ± 6.6 | 15.3 ± 8.2 | 11.1 ± 4.1 | 0.016 |
| RASr (%) | 13.1 ± 8.6 | 15.3 ± 10.5 | 11.3 ± 6.4 | 0.135 |
| DF on surface ECG leads (Hz) | | | | |
| Lead I | 5.7 ± 1.4 | 5.3 ± 1.5 | 6.2 ± 1.3 | 0.009 |
| Lead II | 6.2 ± 1.0 | 6.0 ± 0.9 | 6.5 ± 1.0 | 0.004 |
| Lead III | 6.1 ± 0.9 | 5.8 ± 0.8 | 6.4 ± 1.0 | 0.007 |
| Lead aVR | 5.8 ± 1.0 | 5.6 ± 0.9 | 6.0 ± 1.1 | 0.094 |
| Lead aVL | 6.1 ± 1.2 | 5.8 ± 1.2 | 6.5 ± 1.1 | 0.028 |
| Lead aVF | 6.1 ± 1.1 | 6.0 ± 1.0 | 6.3 ± 1.2 | 0.292 |
| Lead V1 | 6.7 ± 1.2 | 6.4 ± 1.3 | 7.1 ± 0.9 | 0.022 |
| Lead V2 | 6.5 ± 1.2 | 6.2 ± 1.2 | 6.9 ± 1.0 | 0.010 |
| Serum biomarkers | | | | |
| IL-6 (pg/ml) | 2.9 ± 4.4 | 2.2 ± 2.2 | 3.8 ± 5.8 | 0.189 |
| IL-1β (pg/ml) | 6.6 ± 14.5 | 7.5 ± 17.9 | 5.6 ± 10.2 | 0.115 |
| TNF-α (pg/ml) | 24.4 ± 20.1 | 23.3 ± 20.6 | 25.6 ± 19.8 | 0.357 |
| Galectin-3 (ng/ml) | 1.4 ± 1.2 | 1.2 ± 0.6 | 1.6 ± 1.6 | 0.197 |
| High-sensitivity TroponinT (pg/ml) | 9.7 ± 6.5 | 9.2 ± 6.6 | 10.4 ± 6.3 | 0.381 |
| Electromechanical dissociation (Hz) | | | | |
| EMD in TTE | 0.2 ± 1.5 | −0.7 ± 1.2 | 1.2 ± 1.2 | <0.001 |
| EMD in TEE | 0.7 ± 1.2 | 0.2 ± 1.0 | 1.1 ± 1.1 | 0.003 |

Continuous variables are shown as mean ± standard deviation. Categorical variables are shown as n (%). For continuous variables, the unpaired two-sided Student's t test was used to assess differences, after testing for normality with a Shapiro–Wilk's test. Otherwise, the Wilcoxon's rank sum test was used. For categorical variables, the Pearson's chi–squared test was used to assess differences when the expected frequencies were higher than 5. Otherwise, the Fisher's exact test was used. DF values from left precordial leads (V3–V6) were less reliable for estimation of atrial activation rates and were excluded from the analysis.

AF atrial fibrillation, ACEis angiotensin converting enzyme inhibitors, ARBs angiotensin II receptor blockers, DF dominant frequency, EMD electromechanical dissociation, LA left atrium, LASr LA strain during reservoir phase, MRAs mineralocorticoid receptor antagonists, RA right atrium, RASr RA strain during reservoir phase, TIA transient ischemic attack, TEE transesophageal echocardiography, TTE transthoracic echocardiography.

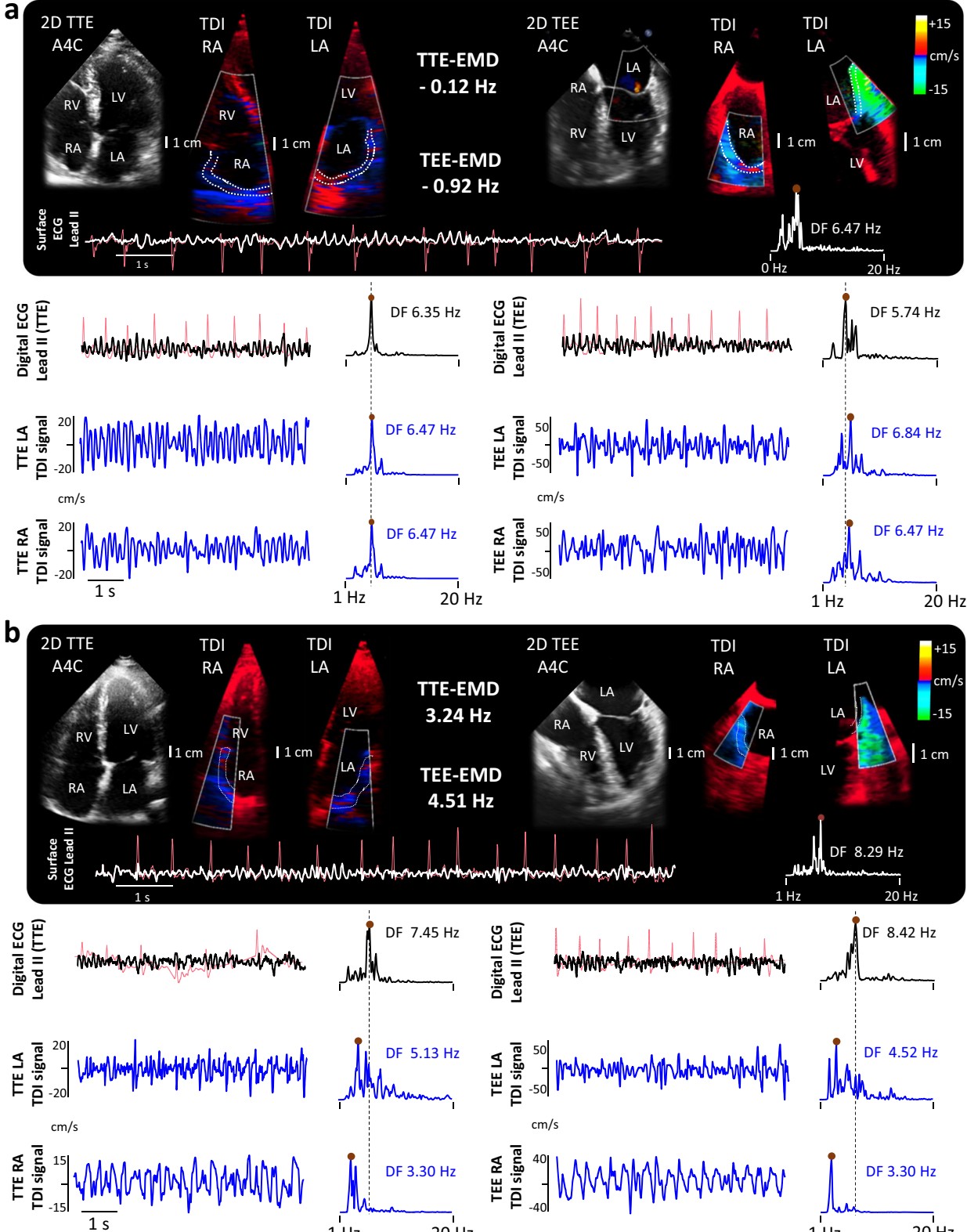

**Fig. 8 | Sample atrial electromechanical assessment in patients with atrial fibrillation. a** Analysis of a 52-year-old man with an atrial fibrillation (AF) episode lasting 61 days. Electromechanical dissociation (EMD) was not documented either on transthoracic echocardiography (TTE) or transesophageal echocardiography (TEE) sequences. Cardioversion to sinus rhythm was documented 12 h after flecainide orally. Representative movies of tissue Doppler imaging (TDI) sequences and atrial wall tracking on the left atrial (LA) wall are shown in Suppl. Movie 1 (TTE

images) and Suppl. Movie 2 (TEE images). **b** Analysis of a 54-year-old woman with an AF episode lasting 153 days. Atrial EMD was detected on TTE and TEE sequences. Pharmacological cardioversion with flecainide orally failed to cardioverter the AF episode within the first 24 h. Light red tracings on digital lead II recordings show subtracted QRS-T complexes. Light red tracings on lead II recordings from the surface ECG show the original signal. A4C apical 4-chamber view, LA left atrium, LV left ventricle, RA Right atrium, RV right ventricle.

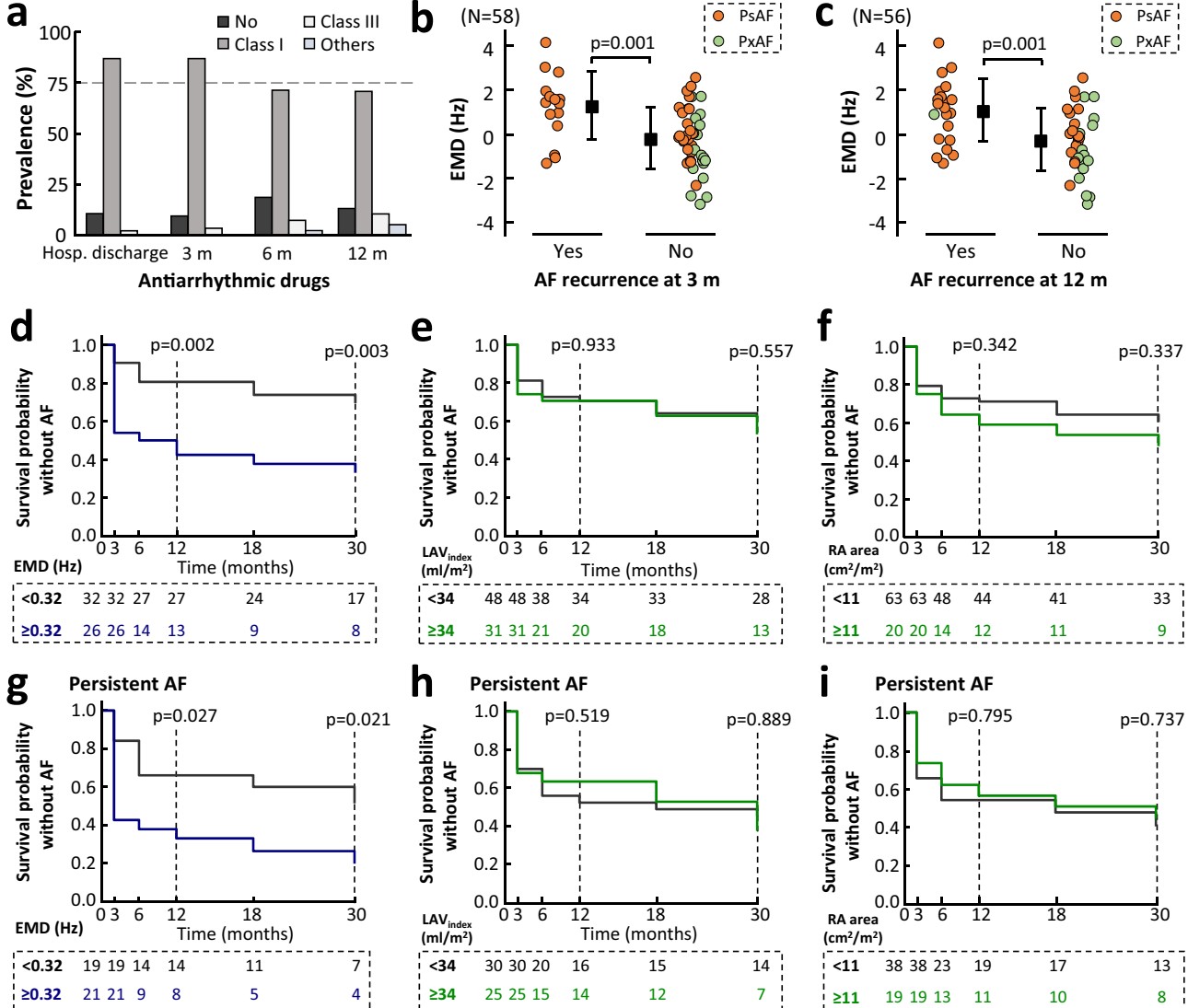

**Fig. 9 | Clinical value of electromechanical assessment in atrial fibrillation recurrences. a** Prevalence of antiarrhythmic drug (AAD) use. **b, c** Comparisons of electromechanical dissociation (EMD) values between patients with and without atrial fibrillation (AF) recurrences at 3 and 12 months of follow-up. The squares and whiskers show mean and standard deviation. Data analysis included the Shapiro–Wilk test for normality followed by unpaired two-sided Student's *t* test to assess differences. **d–f** Kaplan–Meier plots and log-rank test analysis of AF recurrences using the EMD, 3D left atrial volume index ($LAV_{index}$), and right atrial (RA) area (indexed to body surface area) in the entire AF population and in the persistent AF patient subset (**g–i**). In (**d–i**), statistical significance between survival times in each group was assessed by means of log-rank tests at 12 and 30 months. Source data are provided as a Source Data file. Ps/PxAF persistent/paroxysmal AF.

## Discussion

The results of this study demonstrate that atrial EMD is an early indicator of AF remodeling that can be detected using non-invasive electromechanical assessment on TTE studies with simultaneous acquisition of lead II and TDI-derived atrial signals. Moreover, EMD demonstrated early prognostic value for rhythm control in AF patients with episodes lasting ≤6 months, before other echocardiography parameters and blood biomarkers could provide significant value. These results were supported by long-term comprehensive monitoring of AF remodeling in pigs and further invasive electrophysiological studies to demonstrate the presence of EMD in the remodeled atria in vivo. Mechanistic studies in whole heart preparations and single cell simulations demonstrated that AF remodeling decreased the frequency threshold to observe dissociation between transmembrane voltage and intracellular free calcium signals. Further studies in tissue samples and sequential atrial biopsies also revealed underlying atrial damage in the contractile machinery, with the presence of abnormal atrial

cardiomyocyte death and fibrosis that contribute to defective atrial mechanical properties during AF progression.

Atrial mechanical properties during sinus rhythm had been reported as important determinants of major clinical outcomes such as thromboembolic events after cardioversion[8,22]. Thus, patients with atrial stunning after cardioversion are at higher risk of thromboembolic events, especially during the first 7 days following cardioversion[22]. This is consistent with the rate of atrial mechanical recovery after cardioversion[11], which may also take longer in patients with persistent AF episodes lasting several weeks. However, the pattern of atrial mechanical dysfunction or its recovery can be patient-specific[8], an especially important consideration during early remodeling stages when other parameters (e.g., atrial dilation) may still be not present. Moreover, the manifestation of atrial mechanical dysfunction can also be affected by the underlying atrial myopathy[16], among other factors[9].

Despite the undoubted value of knowledge about atrial mechanical activity, assessing wall motion during AF remains challenging. Initial approaches have used tissue velocity imaging at specific atrial

wall spots to assess atrial activation rates during AF[13–15]. Using the diastolic interval between heart cycles, de Vos et al. reported faster tissue velocity imaging waves in 7 patients with permanent AF than in 5 patients with paroxysmal AF[15]. These data correlated well with intracavitary electrogram recordings from a mapping catheter, which the authors attempted to position at a similar location. These results might suggest that information from TDI-waves might be limited to tracking electrical data. However, the differences between our findings and those obtained with previous approaches likely reflect major methodological improvements that enable to demonstrate progressive dissociation of atrial electromechanical activity during AF progression[13–15]. First, we generated specific software tools to process TDI data from the entire atrial surface within the image sector angle, rather than limiting the analysis to a single atrial spot, which could affect cycle length quantification. Second, we used advanced imaging processing and signal quantification for both TDI- and ECG-derived signals rather than relying on pseudomanual quantification of AF cycle length. Third, we obtained a robust measure of EMD from simultaneously acquired TDI and electrical signals, which in pigs were also spatially colocalized with intracavitary recordings. Fourth, we frequently used acquisition rates >300 Hz together with 6-s segments rather than relying on diastolic intervals of a single or a few heart beats with a significantly lower frame rate[15]. Finally, we obtained important insights from mechanistic studies in isolated heart preparations, atrial tissue samples and single cell simulations that provided further evidence of contractile dysfunction and dysregulation of voltage and calcium dynamics during AF. The results are also consistent with previous data reporting a lower pacing frequency threshold for calcium transient alternans in atrial myocytes of patients who develop postoperative AF[16].

Atrial electromechanical assessment during AF represents a significant advance in the definition of objective parameters to assess AF-related remodeling, rather than exclusively relying on temporal criteria[1]. Our results suggest that detecting EMD might be especially relevant during the first months of AF progression, when therapeutic interventions have a greater prognostic impact[23]. Current temporal criteria for defining AF do not consider the underlying broad range of electrical, mechanical, and structural remodeling parameters that vary from patient to patient and can also affect other clinical outcomes (e.g., stroke). Here, we report that electromechanical assessment provides a parameter that can distinguish individual-specific remodeling stages in patients with apparently similar periods in AF. Moreover, in paroxysmal AF, our data suggest that the electromechanical relationship can also be affected by the underlying atrial myopathy (Fig. 7c), resulting in the classification of these patients at more advanced remodeling stages than would be possible with temporal criteria. Thus, similar to electrical remodeling progression[4], electromechanical remodeling need not be at the same stage in all patients in the early weeks and months of AF. Electromechanical information obtained during AF could also be complemented with other imaging tools to assess atrial fibrosis (e.g. magnetic resonance imaging)[24] and function during sinus rhythm (e.g., atrial strain)[25] to more accurately characterize the atrial substrate.

Lead II ECG signals do not provide activation rates from the same atrial region as the image sector angles positioned on the atrial wall during simultaneous TDI acquisitions. However, our experimental data show that DF values from lead II include the activation rate of large endocardial atrial areas (Suppl. Fig. 2). This minimizes the chance of TDI acquisition in atrial areas with much faster or slower EARs than the DF peak detected on lead II-derived atrial ECG. Moreover, data from the 11 animals undergoing simultaneous and colocalized acquisition of endocardial electrograms and TDI data demonstrated that the lead II recordings faithfully detected EMD. Implementation of electromechanical assessment during AF in day-to-day clinical practice will require incorporating new software tools into current echocardiography machines.

## Methods

### Study approval

All animal procedures were approved by the Comunidad de Madrid (Ref# PROEX097/17 & PROEX078.8/21) and conformed to EU Directive 2010/63EU and Recommendation 2007/526/EC regarding the protection of animals used for experimental and other scientific purposes, enforced in Spanish law under Real Decreto 1201/2005. The study in patients was approved by the ethics committees of the Hospital Clínico Universitario San Carlos (Ref#14/273-E) and the Hospital Universitario Central de Asturias (Ref#07/16). The study complies with the Declaration of Helsinki. All patients gave written informed consent.

### Pig model of AF and sham-operated controls

Pigs were housed at the animal facilities of CNIC with adequate and authorized infrastructure to house and care for animals (ISO-9001-N° ES034545). All researchers, technicians and veterinarians involved in the care of the animals and in the experimentation of the study had the necessary accreditations according to Spanish regulations (RD 53/2013). We used slow growing crossbred pigs (Yucatan-Large white) to minimize problems arising from handling over-large animals after long periods with AF. Each was implanted with a dual chamber pacemaker (Accent DR-RF, Abbott) with atrial and ventricular leads inserted into the right atrial appendage and right ventricular apex, respectively. All pigs underwent AV node ablation before attempting HRAP as described elsewhere[3]. All pacemaker implantations were carried out under general anesthesia. Pigs were pre-medicated with a combination of intramuscular Ketamine (20 mg/kg) and Midazolam (0.5 mg/kg). Anesthesia was induced with an intravenous bolus of Fentanyl (0.010 mg/kg). After oral tracheal intubation animals were mechanically ventilated with intermittent positive pressure and anesthesia was maintained by a combination of Fentanyl (0.005 mg/kg/h i.v.) and Sevoflurane (2%). Euthanasia was performed with intravenous administration of sodium pentobarbital (40 mg/Kg).

Thirteen pigs (male/female 77%/23%) were sequentially monitored at 3-week intervals to study atrial remodeling progression during AF. A further 6 pigs (male/female 83%/17%) were used as sham-operated controls and underwent the same follow-up planning although the HRAP protocol was never activated. A further 11 pigs (male/female 50%/50%) with long-term persistent AF (≥ 6 months of self-sustained AF) were used to study simultaneous local electromechanical activity under 3D electroanatomical guidance. A further 16 pigs (male/female 44%/56%) with healthy atria were used to study electromechanical activity during programmed atrial electrical stimulation and short-lasting AF episodes (<3 min duration) after AV node ablation. Animals were ≈6 months old at the time of AV node ablation/AF protocol initiation. Back-up ventricular pacing was set at 40 bpm with a screw-in lead (Tendril STS 52 cm, St. Jude Medical) introduced through the jugular vein and positioned in the right ventricle. All echocardiography studies and ECG recordings required ketamine and midazolam premedication and intubation.

### Tissue samples collection

Atrial biopsies were obtained using a long steerable sheath (8.5 F Agilis NxT Steerable Introducer, 91 cm Lumen Length. Abbott, St. Paul, MN), which was positioned in the right atrial appendage under fluoroscopy guidance. The correct position of the sheath for atrial biopsies was confirmed with a mapping/ablation catheter (Blazer II XP, Boston Sci., Marlborough, MA) to document near field atrial electrograms. Then, a 5.5-French biopsy forceps (Cordis, Miami Lakes, FL) was introduced

through the long sheath to take the atrial endomyocardial biopsy at the preselected atrial location. Tissue samples (3-7 mg) were used for immunoblotting studies, histopathology analysis and further Real Time quantitative PCR for gene expression analysis of lysyl oxidase (See Suppl. Methods and Suppl. Table 2 for expanded details).

At the end of follow-up, after euthanasia, left and right atrial samples and left ventricular samples were also obtained for further immunoblotting analysis, histopathology studies, immunohistochemistry and confocal microscopy imaging (See Suppl. Methods and Suppl. Table 3 for details).

## Proteomic analysis

Quantitative proteomics was performed by multiplexed isobaric labeling. Protein quantification was performed as described elsewhere[26]. One percent false discovery rates was used as criterion for peptide identification. Analysis of quantitative data was performed as described elsewhere[27,28]. Relative protein abundances were expressed as standardized $\log_2$-ratios ($Zq$). Gene Ontology (GO) enrichment was performed from the list of proteins that showed statistically significant changes (493 out of 4920 proteins) over the sequential biopsy studies. GO analysis was carried out with the enrichGO tool in RStudio. GO terms with adjusted $p$-values < 0.05 were considered significant. Among all significant GO terms, we focused the analysis on those related to mechanical and structural remodeling to specifically address proteomic changes potentially affecting the atrial electromechanical relationship (See Suppl. Methods for expanded details).

## In vivo high-density electroanatomical mapping in pigs

The invasive electrophysiology study was performed under general anesthesia (as described above) using percutaneous venous and arterial femoral access to reach the atrial chambers and continuously monitor the blood pressure, respectively. Three-dimensional electroanatomical data and AF maps were obtained using the Ensite Precision system as described in detail elsewhere[3]. Endocardial activation rates were obtained from high-density electroanatomical maps of the left and right atria (Expanded details are provided in the Suppl. Material).

## Simultaneous optical mapping of transmembrane voltage and intracellular calcium transients in the pig heart

A schematic of the optical mapping system developed for this study is shown in Suppl. Fig. 4a. All excitation and emission lights traveled through the same optical fiber (77554; Newport Corporation, Irvine, CA). Four light emitting diodes (LED) light sources (UV LED 1: M340D2; Thorlabs Inc., Newton, NJ; UV LED 2: MTSM385UV-D5120S; Marktech Optoelectronics Inc., Latham, NY; BLUE LED: CBT-90 Blue, RED LED: CBT-90 Red; Luminus Devices Inc., Woburn, MA) were driven by and synchronized with the frame-exposure signal of the emission light sensor (Evolve 128 EMCCD camera; Teledyne Photometrics, Tucson, AZ) using a custom-built circuit. Excitation and dichroic filters were off-the-shelf (EX1: ET340x, EX2: ET380x, EX3: ET480/20x, EX4: ZET635/20x, DC2: ZT349rdc, DC3: T425lpxr; Chroma Technology Corp, Bellows Falls, VT; DC1: Di02-R488-25×36, DC4: NFD01-633-25×36; Semrock Inc., Rochester, NY) and the multi-band emission filter was custom-made (EM: ET525/50 M + 800/200 M; Chroma Technology Corp). Plano-convex lenses were used to collimate the LED light and focus light into the optical fiber and onto the sensor (LA1951; Thorlabs Inc.). The light paths followed by all four excitation lights and the voltage (solid and dashed red arrows; di-4-ANEQ(F)PTEA) and calcium (solid and dashed blue arrows; Fura-2AM) dye emission lights are shown in Suppl. Fig. 4a. The transmission curves of all the excitation, emission and dichroic filters are shown together in Suppl. Fig. 4b.

## Clinical cohort

The clinical cohort consisted of a prospective series of patients admitted to 2 tertiary hospitals from October 2014 to March 2022 with a recent AF diagnosis. The study was designed to include patients <70 years old, at early remodeling stages of AF progression (paroxysmal AF or persistent with ≤6 months of episode duration based on symptoms) and lack of other relevant comorbidities, significant structural heart disease or baseline treatment with antiarrhythmic drugs, other than β-blockers (Fig. 7a, b). More specifically, patients taking amiodarone within the 6-month period before inclusion, or other antiarrhythmic drugs within the month before inclusion, were excluded. All patients were free of heart failure symptoms and other signs of severe cardiac or systemic comorbidities on physical examination, ECG, X-ray, and routine blood testing. After admission, patients with left ventricular ejection fraction <50% or any documented significant valvulopathy (moderate–severe stenosis or regurgitation) were excluded from the study at the time of conventional TTE. We also excluded patients with left atrial volumes >55 ml/m² due to the low probability of effective rhythm control during the follow-up[29]. TEE was performed according to current recommendations to rule out atrial thrombi before attempting cardioversion. Simultaneous TDI and surface lead II recordings were obtained during TTE and TEE (if indicated) to assess the atrial electromechanical relationship. Twelve-lead ECG recordings and blood samples were also obtained before attempting cardioversion with oral flecainide (200 mg or 300 mg for <90 kg or ≥90 kg body, followed by 100 mg every 12 h) during a 24-h window on continuous telemetry. If cardioversion with flecainide failed, electrical cardioversion was performed. Asymptomatic AF recurrences were detected during regular ECG recordings and follow-up visits at 3, 6, 12, 18, and 24 months and annually thereafter. Symptomatic episodes were detected as they occurred.

## Imaging and surface ECG recordings during echocardiography studies

In pigs, left atrial area (indexed to animal weight) was measured in a four-chamber view (Suppl. Fig. 1) by TTE (probe X5-1, Philips) with an iE33 ultrasound system (software-6.0.0.845, Philips healthcare, USA), with the animal in a left lateral decubitus position. TDI sequences were obtained from TEE views (probe X7-2t, Philips) focused on the posterior left atrial wall. In a subgroup of 7 pigs with healthy atria and 8 pigs with long-lasting persistent AF, TDI sequences were also obtained from TEE views focused on the left atrial free wall. We used the best beam alignment, image sector angle, and image depth to enable imaging of the atrial wall at the highest possible frame rate (>200 Hz). TDI and lead II data were acquired simultaneously for 6-s.

In patients, left and right atrial areas were measured at end-systole in the frames just before the opening of the mitral and tricuspid valves, respectively[20]. Three-dimensional atrial volumes were obtained from the apical view using a multibeat full-volume acquisition. The atrial electromechanical relationship was assessed on simultaneous 6-s TDI and lead II ECG segments (frame rate >200 Hz) from TTE and TEE studies performed with either an iE33 ultrasound system (software-6.3.7.745, X5-1 probe) or an Epiq7 system (software-1.8.6, Philips, X7-2t probe). TTE images were acquired with the patient in the left lateral decubitus position, and apical 4-chamber views were obtained with the image sector angle focused and the ultrasound beam aligned on the left and right atrial walls. In TEE studies, the image sector angle was focused on the left atrial wall (between the left atrial appendage and the left pulmonary veins) and the right atrial wall (lateral-roof wall). In patients and pigs, the tissue Doppler scale was set at 15 cm/s to display aliasing-free lower and higher velocities (Fig. 1b and Fig. 8a, b). Further details are provided in the Suppl. Methods.

## Signal processing of ECG and TDI signals

The following TDI sequence data were exported: (i) raw TDI signals from the entire window inside the image sector angle; (ii) superimposed 2D imaging frames to enable visualization of cardiac anatomy; and (iii) simultaneous lead II recordings (Suppl. Fig. 11, 12). All

data were processed using custom-made software in Matlab (R2016b) (See Suppl. Methods for expanded details). Briefly, atrial wall segmentation and tracking over the 6-s acquisition segments was based on manual selection of the atrial region of interest (within the image sector angle) and echogenicity criteria on superimposed 2D images (Fig. 1b and Suppl. Movie 3). TDI signals within the atrial wall were processed using the empirical mode decomposition method (Suppl. Fig. 13–15) to minimize variable ventricular motion effects on atrial TDI signals. This method was more effective than principal component analysis in the presence of time-varying ventricular motion artifacts caused by irregular diastolic intervals in AF (Suppl. Figs. 16, 17)[30]. Briefly, empirical mode decomposition uses the original Doppler signal in the time domain to obtain intrinsic oscillatory modes (intrinsic mode functions; IMFs) without assuming periodicity. The method provides fully data-driven, unsupervised signal decomposition without assumptions about the data. The method also satisfies the perfect reconstruction property so that superimposing all extracted IMFs together with the residual slow trend reconstructs the original signal without information loss or distortion (Suppl. Figs. 13, 14). We consistently used IMFs within the physiological range of atrial activation rates during AF (IMFs 4, 5, 6 + / −3, 7), excluding very low frequency waves (ventricular motion, if present; Suppl. Fig. 12, 13) and very high frequency waves (potential noise, Fig. 1c, e). TDI-derived MAR was then calculated using spectral analysis and the fast Fourier transform to convert the tracing into the frequency domain (power spectral density). The DF was used to obtain a rapid calculation of the reciprocal average MAR (Fig. 1c, e). The mean amplitude excursion of TDI signals was also calculated (see Suppl. Methods and Suppl. Fig. 9 for details).

The atrial frequency content of simultaneous lead II tracings from TDI sequences and digitized 12-lead ECG tracings was obtained after removing ventricular components (QRS-T complexes) using principal component analysis (Suppl. Fig. 2), as described elsewhere[3]. Atrial electrical activation frequency was then calculated using the fast Fourier transform to convert the atrial tracings into the frequency domain and obtain the DF (Suppl. Fig. 11)[3]. Further details are provided in the Suppl. Methods.

### Definition of atrial electromechanical dissociation
Atrial EMD was defined as a faster atrial EAR than the simultaneous counterpart TDI-derived MAR (EMD = EAR-MAR). In patients, left and right atrial wall data were included in EMD assessment as follows:

$$\text{EMD} = \frac{\text{EMD}_{\text{right atrium(RA)}} + \text{EMD}_{\text{left atrium(LA)}}}{2} = \frac{(\text{EAR} - \text{MAR}_{\text{RA}}) + (\text{EAR} - \text{MAR}_{\text{LA}})}{2}$$

### Analysis of circulating biomarkers
In patients, plasma levels of galectin-3, interleukin1-β, tumor necrosis factor-α (TNF- α), and interleukin-6 were determined by high-sensitivity ELISA with commercial kits (Human ELH-Galectin-3 and Human Interleukin1-β from Raybiotech, Norcross, GA; human high-sensitivity TNF-α immunoassay from R&D Biosystems, Minneapolis, MN; human high-sensitivity Interleukin-6 from Besançon Cedex, France). ELISA sensitivity was 0.6 ng/ml, 0.3 pg/ml, 1.6 pg/ml, and 0.8 pg/ml for galectin-3, interleukin1-ß, TNF- α, and interleukin-6, respectively. The intra- and inter-assay variation coefficients were <10% and 12%, for both galectin-3 and interleukin1-β; <4% and 9% for interleukin-6; and <5% and 6% for TNF- α. Hs-cTnT was analyzed using an electrochemoluminometric assay in a Cobas e601 platform (Roche-Diagnostics, Basel, Switzerland).

In pigs, plasma PIIINP levels were determined by immunoassay using high-sensitivity ELISA commercial kits (Pig PIIINP–Ref. MBS742383, MyBiosource, San Diego, CA), according to manufacturer's specifications.

### Statistical analysis
Continuous variables are reported as median and interquartile range except where noted. Data normality was assessed with the Shapiro–Wilk test. Linear correlation between variables was performed using the Pearson or Spearman correlation coefficient, as necessary. Statistical significance was assessed by the T-test or the Mann–Whitney/Wilcoxon test, as appropriate. Categorical variables and percentile comparisons were compared using a Chi−squared test or the Fisher exact test, as required. A two-way repeated measures ANOVA was used for follow-up comparisons between AF animals and controls. In patients, non-correlated variables (correlation coefficients <0.8) among the statistically significant variables in the univariate analysis, as well as those with $p \leq 0.2$, were used to generate four predictive models of the primary outcome using a logistic regression approach. This approach aimed to provide the individual value of an objective and non-invasive EMD-based model (using TTE) compared with an ECG-based electrical model or more conventional models based on clinical variables or AF classification. Clinical and ECG-based multivariate models were generated using all possible combinations up to 2 variables to avoid overfitting, based on the number of events for each variable included in the model[31]. Mean imputation was used for missing data in predictive models, and only for variables with a number of missing values ≤ 10 and normal distribution. The best predictive models were preselected based on the Akaike information criterion[32]. The final clinical and ECG-based models were then selected based on the AUC of receiver operating characteristic curves. The increased certainty provided by each model when added to AF classification was assessed by integrated discrimination improvement[33]. Kaplan–Meier plots were used to depict time-to-AF recurrence, together with Mantel-Cox paired comparisons (log-rank test). Differences were considered statistically significant at $p < 0.05$. All analyses were performed with Stata/IC 15.1 and RStudio (3.5.2) for graphical representations.

### Reporting summary
Further information on research design is available in the Nature Portfolio Reporting Summary linked to this article.

## Data availability
All custom-written code, graphical user interfaces, Stata scripts for logistic models and examples of test data are available at https://gitlab.com/Advanced_Development_in_Arrhythmia_Mechanisms_and_Therapy_Lab/Electromechanical_Assessment_During_AF. All data needed to reproduce the study can be found in the manuscript, supplementary information and source data. Individual patient data cannot be shared without permission of the study participants and ethical approval. The mass spectrometry proteomics data are available via ProteomeXchange with identifier PXD043016. Source data are provided with this paper.

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

## Acknowledgements

This work was supported by the European Union Horizon 2020 research and innovation program under Grant Agreement#965286. The study was also supported by the Ministry of Science and Innovation (MCIN) (PID2019-109329RB-I00 and PGC2018-097019-B-I00) funded by MCIN / AEI / 10.13039/501100011033, the Instituto de Salud Carlos III (Fondo de Investigación Sanitaria grant PRB3) (PT17/0019/0003- ISCIII-SGEFI / ERDF, ProteoRed), the Fundación Interhospitalaria para la Investigación Cardiovascular, the Fundación Salud 2000 and by "la Caixa" Banking Foundation (project code HR17-00247). The Centro Nacional de Investigaciones Cardiovasculares (CNIC) is supported by the Instituto de Salud Carlos III (ISCIII), the Ministerio de Ciencia e Innovación (MCIN) and the Pro CNIC Foundation, and is a Severo Ochoa Center of Excellence (CEX2020-001041-S funded by MICIN/AEI/10.13039/501100011033). We also thank Asunción Conde and Sergey Mironov for their support on monitoring database quality and advice in ECG signal processing, respectively.

## Author contributions

Conceptualization: D.F.R., J.G.Q.. Methodology: D.F.R., J.G.Q., D.E.V., M.C.S., A.S.C., J.O.L., A.G.O., A.G.E., J.S.G., J.M.A.A., D.G.G., P.L., L.L., J.V., N.P.C., J.P.V., M.J.G.T.. Investigation: D.E.V., A.G.E., J.G.Q., J.M.A.A., P.M., J.M., A.O.H., P.M.A., R.M., E.C., R.G.G., P.L., L.L., P.Y., A.S.C., G.L.R., A.R.R., J.B.M., C.N.P.G., J.M.S., D.C., J.M.D.L.H., J.S.G., D.G.G., M.C.S.. Visualization: D.F.R., D.E.V., A.G.E., J.G.Q., P.L., A.S.C., G.L.R., J.S.G., J.M.A.A., R.M., P.M., P.M.A., D.C., J.M.D.L.H., M.J.G.T., D.G.G.. Patient recruitment: D.F.R., D.E.V., M.J.G.T., J.P.V., N.P.C., C.N.P.G., J.M.S., J.B.M., D.C.. Funding acquisition: D.F.R., J.P.V., J.V.. Project administration: D.F.R., M.J.G.T..

Study design: D.F.R.. Supervision: D.F.R., J.P.V., M.J.G.T., N.P.C.. Writing – original draft: D.F.R., D.E.V., J.G.Q.. Writing – review & editing: All authors.

## Competing interests

L.M.L. and P.Y. are founders and owners of Potentiometric Probes LLC, which sells voltage sensitive dyes. P.L. is both an owner and employee of Essel Research and Development Inc. D.F.R. and J.G.Q. are coinventors on a pending patent application pertaining to the methodology to assess electromechanical activity during cardiac fibrillation. The remaining authors declare no competing interests.

## Additional information

Daniel Enríquez-Vázquez[1,2,3,18], Jorge G. Quintanilla[1,3,4,18], Alba García-Escolano[1,5,18], Marinela Couselo-Seijas[1], Ana Simón-Chica[1], Peter Lee[6], José Manuel Alfonso-Almazán[1], Patricia Mahía[4], Andrés Redondo-Rodríguez[1], Javier Modrego[1,3,7], Adriana Ortega-Hernández[7], Pedro Marcos-Alberca[4], Ricardo Magni[1], Enrique Calvo[1,3], Rubén Gómez-Gordo[7], Ping Yan[8], Giulio La Rosa[1], José Bustamante-Madrión[9], Carlos Nicolás Pérez-García[1,4], F. Javier Martín-Sánchez[9], David Calvo[10], Jesús M. de la Hera[10], María Jesús García-Torrent[11], Álvaro García-Osuna[12,13], Jordi Ordonez-Llanos[12,14,15], Jesús Vázquez[1,3], Julián Pérez-Villacastín[3,4,16], Nicasio Pérez-Castellano[3,4,16], Leslie M. Loew[8], Javier Sánchez-González[17], Dulcenombre Gómez-Garre[3,7] & David Filgueiras-Rama[1,3,4] ✉

[1]Novel Arrhythmogenic Mechanisms Program, Centro Nacional de Investigaciones Cardiovasculares (CNIC), Madrid, Spain. [2]Servicio de Cardiología, Instituto de Investigación Biomédica A Coruña (INIBIC), Complexo Hospitalario Universitario A Coruña, A Coruña, Spain. [3]Centro de Investigación Biomédica en Red de Enfermedades Cardiovasculares (CIBERCV), Madrid, Spain. [4]Cardiovascular Institute, Instituto de Investigación Sanitaria del Hospital Clínico San Carlos (IdISSC), Madrid, Spain. [5]ETSI Telecomunicación, Universidad Politécnica de Madrid, Madrid, Spain. [6]Essel Research and Development Inc., Toronto, ON, Canada. [7]Laboratorio de Microbiota y Biología Vascular, Instituto de Investigación Sanitaria del Hospital Clínico San Carlos (IdISSC), Madrid, Spain. [8]Richard D. Berlin Center for Cell Analysis and Modeling, University of Connecticut School of Medicine, Farmington, CT, USA. [9]Emergency Department, Instituto de Investigación Sanitaria del Hospital Clínico San Carlos (IdISSC), Madrid, Spain. [10]Hospital Universitario Central de Asturias, Instituto de Investigación Sanitaria del Principado de Asturias, Oviedo, Spain. [11]Department of Medicine, Universidad Complutense de Madrid, Madrid, Spain. [12]Department of Clinical Biochemistry, Hospital de la Santa Creu i Sant Pau, Barcelona, Spain. [13]Institut de Recerca de l'Hospital Santa Creu i Sant Pau, Institut d'Investigacions Biomèdiques, IIB Sant Pau, Barcelona, Spain. [14]Universidad Autónoma, Barcelona, Spain. [15]Foundation for Clinical Biochemistry & Molecular Pathology, Madrid, Spain. [16]Fundación Interhospitalaria para la Investigación Cardiovascular (FIC), Madrid, Spain. [17]Philips Healthcare Iberia, Madrid, Spain. [18]These authors contributed equally: Daniel Enríquez-Vázquez, Jorge G. Quintanilla, Alba García-Escolano. ✉e-mail: david.filgueiras@cnic.es

