## [Peer Review File · Nature Communications]

Non-invasive electromechanical assessment during atrial fibrillation identifies underlying atrial myopathy alterations with early prognostic valueREVIEWER COMMENTS

Reviewer #1 (Remarks to the Author):

This is a very well-conducted study and the authors have done an excellent job in preparing the paper. I have few comments. Please address those pointwise.

1. All patients remained on AAD for 1 year (mostly class I). How did the authors exclude the proarrhythmic (EMD) effect of these drugs? Please discuss and provide data if available.
2. The algorithm to assess EMD seems quite complex. How feasible it is to implement this complex analysis in the day-to-day clinical practice? Please discuss it as a limitation
3. The paper is very long with too much information. Therefore, the authors should consider providing the main results (both preclinical and clinical) as bullet points in a box for the convenience of the readers.
4. Please expand MAR and DF. Please also consider using the fully expanded terms instead of the abbreviations. It is tough to keep track as it is.
5. Figures are too busy. Can you please separate those instead of giving so many subheadings under one figure?

Reviewer #2 (Remarks to the Author):

This manuscript reported a non-invasive electromechanical assessment using Tissue Doppler Imaging measurement of atrial electromechanical dissociation as a prognostic indicator for a successful acute pharmacological cardioversion. They used a pig model of long-term atrial fibrillation and performed longitudinal electromechanical assessment using lead II-derived atrial ECG traces and simultaneous TDI-derived signals from TEE or TTE. Atrial biopsies were done during AVN ablation, 6 week (paroxysmal Af period) and during late persistent Af period. They applied various electrophysiological methods, including optimal mapping of simultaneous voltage and intracellular calcium (both using fluorescent methods) in isolated Langendorff-perfused hearts. Although the electrophysiological methods are robust, lack of molecular mechanism and further exploration of other potential molecular biomarkers substantially reduce enthusiasm of this article.

Major points:

1. Were all of these measurements performed in the same areas of atria? How did the authors control for heterogeneity within atria? Did they focus on the areas adjacent to pulmonary veins?
2. The EMD threshold of 0.32 Hz was not strongly justified and seemed arbitrary.
3. What about the magnitude / amplitude of TDI signal, which may represent "atrial contraction" or other strain-related parameters? These potential new parameters should also be considered in addition to EMD.
4. Any data on TDI obtained from TEE vs TTE in Af patients?
5. "Usefulness of tissue Doppler-derived atrial electromechanical delay for identifying patients with paroxysmal atrial fibrillation" by Kanako Akamatsu, et al in Cardiovascular Ultrasound volume 18, Article number: 22 (2020) reported that the lateral EMD and the left atrial volume index were significant independent predictors for PAF patients. This paper reported a similar, though much more robust, data supporting the use of EMD as predictor of successful pharmacological cardioversion (AUC=0.874). This incremental knowledge should be expanded by other novel biomarkers and molecular mechanisms.
6. Fig.3E showed that TDI signal significantly dropped below lead II atrial signal (suggesting EMD) starting from 9 weeks, while the proteomics data showed that proteins related to muscle contraction dropped starting from 6w. I believe that this implies that molecular biomarkers precede EMD. Further exploration of underlying mechanisms and molecular biomarkers will significantly increase enthusiasm. Confirmatory analysis by immunostaining and Western blots are also required.
7. Overall, the manuscript is very hard to read and will need re-writing in many parts.

Minor points:

1. Line 131: average of 376.4 +/- 111.7 days and 702.9 +/- 189.2 days show a huge variability in the timing. This needs justification and explanation.
2. Fig. 4 is very confusing; Any data of AF without remodeling in Fig.4E?

Reviewer #3 (Remarks to the Author):

Manuscript Titled, Non-invasive electromechanical assessment during atrial fibrillation identifies 2 underlying atrial myopathy alterations with early prognostic value, by Enríquez-Vázquez, etc. presents a critical parameter, Atrial electromechanical dissociation (EMD) in the prognostic prediction of atrial fibrillation (AF) development. The study developed a method of EMD calculation and evaluated it in animals (preclinical) with and without AF and even in the clinical cohort, which showed interesting results and significance. The best part of the study is that it suggested EMD is earlier than biomarkers in atrial remodeling and AF prediction, I have a few comments:

- Regarding the DF of EMD, how to explain that, in paroxysmal AF, DF is -0.61Hz?
- If the EMD could be done at the segmentation level, the distribution of EMD could provide tips for AF ablation. Is there any difference in the atrial distribution of EMD?
- The study showed atrial area increased during atrial remodeling as usual. How the size of the area affects the EMD value? As we know, in clinical scenarios, some AF patients have average atrial size.
- How to explain, in PsAF remodeling heart, the voltage-calcium dissociation is not available in high frequency.

Reviewer #4 (Remarks to the Author):

This is an interesting and very comprehensive study that addresses several aspects of electromechanical characterization during atrial fibrillation (AF). This study reports novel mechanistic insights into the electromechanical remodeling process associated with AF progression and translates its prognostic value in the clinic. The study provides experimental support from a pig model in which sequential electromechanical assessment during AF progression showed a progressive decrease in mechanical activity and early dissociation from its electrical counterpart during in vivo studies. Novel optical mapping techniques applied to the contracting atria of Langendorff-perfused hearts during AF demonstrated underlying dissociation of transmembrane voltage and intracellular calcium transients supporting the in vivo results. Atrial biopsies at different stages of AF-related atrial myopathy further revealed early downregulation of contractile proteins and progressive increase in biological functions associated with atrial fibrosis. In patients, non-invasive assessment of the atrial electromechanical relationship during AF enabled novel characterization of the underlying atrial myopathy at early stages of AF progression. Atrial electromechanical dissociation was a prognostic indicator for acute pharmacological cardioversion, and also performed better than other conventional cardiac imaging parameters to early predict AF recurrences during the follow-up.

This study provides several significant novel findings that include identifying Electro mechanical dissociation as an early marker of atrial remodelling progression during AF, novel optical mapping ion channel dynamics and validated TDI in a clinical setting.

I have minor comments on this excellent manuscript.

- 1) Electromechanical assessment in the pig model: Please provide more details on how you defined short lasting AF in this model i.e. < 5 minutes etc.
- 2) Was the threshold to achieve EAR steady state affected by burden of AF preceding this finding?
- 3) Was there a direct correlation between LA area and EARs?
- 4) Can serial pharmacological cardioversions prevent reaching the plateau? In other words can interventions prevent AF from progressing to persistent or complete electrical remodelling?
- 5) There was 1 pig that never achieved the steady state what protected this pig, there may be some interesting insights derived from this pig that may provide information on a protective innate mechanisms that allows some to never progress to persistent AF. I understand this is only 1 animal but important information may be obtained from this finding.
- 6) Calcium transient alternans: Can you indicate what % of the hearts showed dissociation that was preceded by calcium transient alternates in the text sentence 229.
- 7) Can you provide information whether cardioversions were performed with or without antiarrhythmic drug therapy i.e. amiodarone?
- 8) Acute cardio version is an important endpoint, however any data on long-term maintenance of sinus rhythm?

9) Discussion: It may be possible that the biomarkers measured did not provide early evidence or predicted EMD, i.e. BNP. Newer biomarkers may be more relevant.

Reviewers' comments

Reviewer #1 (Remarks to the Author):

This is a very well-conducted study and the authors have done an excellent job in preparing the paper. I have few comments. Please address those pointwise.

We thank the reviewer for such encouraging comments about the study. We have further improved the study with extensive work aiming to address all reviewers' comments.

1. All patients remained on AAD for 1 year (mostly class I). How did the authors exclude the proarrhythmic (EMD) effect of these drugs? Please discuss and provide data if available.

After cardioversion (pharmacological or electrical cardioversion; the latter only in case of non-successful pharmacological cardioversion with flecainide during the first 24 h) all patients underwent an initial rhythm control strategy which was mainly associated with Class I antiarrhythmic drugs (new Figure 9A). After 1 year of follow up, 86.6% of patients remained with antiarrhythmic drugs (Class I 70.6%, Class III 10.6%, and others 5.3%). However, none of patients were taking antiarrhythmic drugs before the inclusion in the study at the time of electromechanical assessment. This is an important comment, since antiarrhythmic drugs might have potentially affected electromechanical assessment. To clarify this issue, in the Results we had mentioned that “Patients under baseline treatment with antiarrhythmic drugs (including calcium channel blockers and digoxin) other than β -blockers were excluded”. In the new version of the manuscript, we have included more specific details in the Methods to address this comment: “More specifically, patients taking amiodarone within the 6-month period before inclusion, or other antiarrhythmic drugs within the month before inclusion, were excluded” (Page 16, line 32; Page 17, lines 1, 2).

Therefore, antiarrhythmic drugs did not have any effect on electromechanical assessment in the patient population included in the study. In fact, electromechanical assessment was used as a prognostic parameter before flecainide administration. During the follow-up, baseline electromechanical assessment may have an important role on early identifying patients with higher risk of recurrences despite an active rhythm control strategy with antiarrhythmic drugs during the follow up. Consistent with this argument, in the study, AF patients with electromechanical dissociation values ≥ 0.32 Hz showed a statistically significant association with the subsequent need of ≥ 1 antiarrhythmic drug and AF ablation during the follow-up (New Suppl. Figure 10A).

2. The algorithm to assess EMD seems quite complex. How feasible it is to implement this complex analysis in the day-to-day clinical practice? Please discuss it as a limitation.

We agree that our methodology to assess atrial electromechanical activity is not clinically available in current echocardiography machines. However, simultaneous acquisition of single lead ECG recordings and Tissue Doppler data at fast acquisition rates are common features of conventional echocardiography machines. Therefore, it sounds reasonable that leading companies in echocardiography can easily implement this type of analysis with user-friendly software tools. In any case, this is obviously beyond are capabilities, although we think it will be a highly clinically relevant tool to improve atrial remodeling characterization in day-to-day clinical practice.

We have included a new sentence in the Limitations to further address this comment: “Implementation of electromechanical assessment during AF in day-to-day clinical practice will require incorporating new software tools into current echocardiography machines.” (Page 14, lines 3-5).

3. The paper is very long with too much information. Therefore, the authors should consider

providing the main results (both preclinical and clinical) as bullet points in a box for the convenience of the readers.

The manuscript includes an extensive and comprehensive number of experimental settings complemented with clinical results. Such translational approach on a novel diagnostic and prognostic tool required quite a long manuscript to demonstrate the scientific rigor we have put into the study. Notwithstanding, we think this comment is important to summarize the main findings of the study for readers with limited time to go into specific details of the methods and results. In the first paragraph of the discussion (Page 11, lines 27-31; Page 12, lines 1-8), we have included a brief summary of the main results and their potential clinical value.

4. Please expand MAR and DF. Please also consider using the fully expanded terms instead of the abbreviations. It is tough to keep track as it is.

MAR (mechanical activation rates) and DF (dominant frequency) are mentioned 35 times throughout the main manuscript (excluding figure legends). DF is also used in 5 of 9 Figures. MAR is also used in the formula to define atrial electromechanical dissociation. The latter makes quite difficult to expand only MAR without doing the same for the other abbreviations present in the formula. Eliminating these 2 abbreviations would also increase the number of words in an already quite long manuscript, in which we have incorporated new data to address all reviewers' concerns. However, we have reduced the number of abbreviations in the main text for other common terms like left atrium, left ventricle, etc, which have not been abbreviated to facilitate the reading.

Therefore, after considering the pros and cons of this comment we have decided to keep these 2 abbreviations, which we think provides the most efficient approach for understanding the entire manuscript as a whole.

5. Figures are too busy. Can you please separate those instead of giving so many subheadings under one figure?

We have reorganized the figures to make them easier to follow. We have separated data from 3 Figures to generate a new Figure 4 (complemented with new experiments to validate proteomics analysis), and two Suppl. Figures 6 and 10. The latter enabled us to keep in the main manuscript more concise versions of new Figure 6 and Figure 9. A new Figure 5 with new data requested from the reviewers has been also included with the most relevant information about biomarkers and underlying atrial cardiac damage during AF. The rest of complementary analyses requested from the reviewers have been included in the Suppl. Material and Suppl. Figures.

Reviewer #2 (Remarks to the Author):

This manuscript reported a non-invasive electromechanical assessment using Tissue Doppler Imaging measurement of atrial electromechanical dissociation as a prognostic indicator for a successful acute pharmacological cardioversion. They used a pig model of long-term atrial fibrillation and performed longitudinal electromechanical assessment using lead II-derived atrial ECG traces and simultaneous TDI-derived signals from TEE or TTE. Atrial biopsies were done during AVN ablation, 6 week (paroxysmal AF period) and during late persistent AF period. They applied various electrophysiological methods, including optical mapping of simultaneous voltage and intracellular calcium (both using fluorescent methods) in isolated Langendorff-perfused hearts. Although the electrophysiological methods are robust, lack of molecular mechanism and further exploration of other potential molecular biomarkers substantially reduce enthusiasm of this article.

We thank the reviewer for highlighting the robustness of the electrophysiological methods we have used to demonstrate the clinical value of this novel approach to simultaneously assess electromechanical activity during atrial fibrillation. We also thank the specific comments to improve the experimental part

and scientific value with the exploration of molecular analysis and biomarkers. In the new version of the manuscript, we have included a substantial number of new *in vivo* experiments in pigs to further study regional differences in atrial electromechanical activity. We have also analyzed additional parameters as the amplitude of the tissue Doppler signals and the atrial longitudinal strain. We have dedicated special effort to the identification of atrial damage and its expression in biomarkers that may indicate the underlying atrial myopathy. Further protein analyses have been also done in atrial and available biopsy samples to support the results from the proteomics. The latter was complemented with single cell simulations to further support the experimental results. Overall, we think the new version of the manuscript represents a multidisciplinary and translational work with highly novel insights into atrial electromechanical remodeling and its impact in clinical outcomes at early stages of AF progression. The specific response for each of the questions is provided below.

Major points:

1. Were all of these measurements performed in the same areas of atria? How did the authors control for heterogeneity within atria? Did they focus on the areas adjacent to pulmonary veins?

In animals (with short-lasting AF episodes, Figure 1 data; or with long-lasting persistent AF, Figure 2 data) tissue Doppler imaging (TDI) sequences using transesophageal echocardiography views were focused primarily on the posterior left atrial wall. In the pig, this region is in close proximity to a large pulmonary vein trunk. In the initial set of experiments performed in healthy animals with short-lasting AF episodes (N=9), all TDI sequences had been acquired from this area of the posterior left atrium. To address any potential heterogeneity on atrial electromechanical activity from different atrial regions in the healthy atria during short-lasting AF episodes, we have included a new set of 7 more animals in which TDI sequences were acquired with the echocardiographic view focused first on the left atrial free wall and then on the posterior left atrium. Those sequences were obtained with simultaneous acquisition of surface ECG recordings from lead II to assess electromechanical activity. The new data are shown in Figure 1G, H (see also below) where neither the posterior left atrial wall nor the left atrial free wall showed relevant differences between electrical and mechanical activation rates during short-lasting AF episodes in pigs with healthy atria. These results have been also included in the main text (Page 4, lines 29-32).

In animals with long-lasting persistent AF, the transesophageal echocardiography views were focused primarily on the posterior left atrial wall. However, in 7 of 10 animals we were also able to obtain TDI sequences from the left atrial free wall, although the initial analysis had been done only with the sequences from the posterior left atrium to keep consistency. In the new version of the manuscript, we have included one more animal with long-lasting persistent AF, which enabled us to have 8 animals with TDI sequences from the posterior left atrium and the left atrial free wall. Those sequences were obtained with simultaneous acquisition of surface ECG recordings to assess the electromechanical remodeling. Moreover, in persistent AF animals, all TDI sequences also included simultaneous data from intracardiac electrograms on the same atrial regions

where the echocardiographic view was focused on (Figure 2B, E). The new analyses are shown in Figure 2D-F and in the main manuscript text (Page 5, lines 14-16). Unlike regional electromechanical assessment in animals with healthy atria during short-lasting AF episodes (Figure 1G, H, above), in animals with long-lasting persistent AF and TDI images from the left atrial free wall and posterior left atrium, there was overt dissociation between electrical and mechanical activation rates in all movies at any of these to two locations (See new Figure 2E on the right).

Overall, the data in the animal model support that atrial mechanical changes may affect the entire left atrium. However, the differences between electrical and mechanical activation rates will be larger in atrial regions with faster electrical activation rates as shown in Figure 2D, in which the data from intracardiac electrograms was obtained at the same location of the TDI signals (see also Figure 2B).

In patients, TDI sequences included left and right atrial wall data to assess electromechanical assessment. In fact, electromechanical dissociation (EMD) was calculated as follows:

$$EMD = \frac{EMD_{right\ atrium\ (RA)} + EMD_{left\ atrium\ (LA)}}{2} = \frac{(EAR - MAR_{RA}) + (EAR - MAR_{LA})}{2}$$

Where, EAR: electrical activation rates; MAR: mechanical activation rates; RA: right atrium; LA: left atrium.

Therefore, the results from patients already considered a more global assessment of the electromechanical remodeling either using transthoracic or transesophageal echocardiography imaging. This approach was not possible in a consistent manner in pigs, since transesophageal echocardiography windows in pigs make it very challenging to obtain right atrial images. Importantly, for electromechanical assessment in pigs, we used transesophageal echocardiography views because the transthoracic windows did not enable us an optimal apical four chamber view with appropriate tissue Doppler beam alignment.

Specific subanalysis of the right and left atrial data from patients shows that the right atrium was more sensitive to dissociation. Based on this comment, we specifically analyzed electromechanical data from the right and left atria independently. Interestingly the right atrium seems to be more sensitive to electromechanical dissociation at electrical activation rates <6.5 Hz. Please, see this analysis on the right. We have not included this subanalysis in the main manuscript since electromechanical assessment, including both left and right TDI atrial data, provides the best predictive model for the primary outcome and AF recurrences at 1 year of follow-up (AUC 0.875).

2. The EMD threshold of 0.32 Hz was not strongly justified and seemed arbitrary.

We are sorry that the 0.32 Hz threshold was not explained well enough in the previous version of the manuscript. We did not provide a strong justification for it since it involved several signal theory concepts we felt might be excessive for the potential readers of the manuscript. In the new version of the manuscript, we provide all the details needed to justify such a threshold. We have changed the references to such a threshold in the main manuscript for clarifying them. We have also included a new section and an additional Supplemental Figure in the Supplementary Material. Please, see the new section “*Spectral resolution and frequency threshold for detecting electromechanical dissociation*” for details (Suppl. Material Page 9, 10) and new Suppl. Figure 18 (on the right). We hope that these detailed explanations may help to understand the rationale behind the 0.32 Hz threshold.

Briefly, two sinusoidal signals, $\cos(2\pi f_1 t)$ and $\cos(2\pi(f_1 + \Delta f)t)$, with a sample rate of at least 200 Hz (the minimum requested for our acquisitions) and slightly different frequencies (f_1 and $f_1 + \Delta f$) would not be discernible in the frequency spectrum and would be displayed as a single fused spectral lobe unless $\Delta f > 0.32$ Hz. Therefore, the minimal Δf above which two distinct spectral lobes start to be discernible enough (i.e., spectral resolution) was 0.32 Hz with the power spectral density estimation method we used for signal processing (See details in the Suppl. Material, Page 9, 10). Therefore, we set 0.32 Hz as the minimal difference between the dominant frequency (DF) peaks of simultaneous ECG and TDI signals for considering that electromechanical dissociation (EMD) was actually present.

3. What about the magnitude / amplitude of TDI signal, which may represent “atrial contraction” or other strain-related parameters? These potential new parameters should also be considered in addition to EMD.

This is a relevant question. We have done new and comprehensive analyses to address this question. Our testing hypothesis was that neither tissue Doppler amplitude nor atrial strain parameters conventionally used in the clinic will be able to provide the same information as the activation frequency of tissue Doppler signals.

More specifically, the amplitude of tissue Doppler signals may be affected by other factors as the myocardial mass, which may vary among patients. This effect would be similar to voltage amplitude differences observed on the QRS complex of patients with left ventricular hypertrophy compared to a normal heart. In any case, the amplitude of the tissue Doppler signal would not provide information about tissue activation rates. To address this concern, in the new version of the manuscript we have measured the mean amplitude excursion of TDI signals to study any potential association between this parameter and the primary outcome (i.e., pharmacological cardioversion within the first 24 hours after flecainide orally). For each TDI signal we quantified the amplitude excursion of both positive and negative deflections and the average value of such excursions (mean Δ TDI). Interestingly, unlike the electromechanical dissociation quantification, Δ TDI in both right and left atria did not differ significantly between patients with successful and non-successful pharmacological cardioversion. We have included this parameter and comparisons in Table 1 and Page 10 (lines 11, 12) of the main manuscript. More detailed information about the methodology to quantify tissue Doppler amplitude and

representative sample cases are also shown in the Suppl. Material (Page 9, 10) and new Suppl. Figure 9 (see also below).

Suppl. Figure 9. Quantification of amplitude excursion of tissue Doppler imaging signals and prognostic value for the primary outcome. **A**, Amplitude quantification includes the positive and negative deflections of the tissue Doppler imaging (TDI) signals (in blue). Then, the average value of all amplitude excursions was calculated for each 6-second TDI segment (mean Δ TDI). **B**, Mean Δ TDI in the left atrium (LA) did not differ significantly between patients with successful and non-successful pharmacological cardioversion (PhCV) within 24 hours after flecainide orally. **C**, Examples of TDI signals from the LA with low/high mean Δ TDI and successful/non-successful PhCV. Unlike electromechanical dissociation values (EMD), the amplitude excursion of TDI signals in the LA was not statistically associated with the primary outcome (PhCV). **D**, Mean Δ TDI in the right atrium (RA) did not differ significantly between patients with successful and non-successful PhCV. **E**, Examples of TDI signals from the RA with low/high mean Δ TDI and successful/non-successful PhCV. Similar to the LA analysis, the amplitude excursion of TDI signals in the RA was not statistically associated with the primary outcome (PhCV within 24 hours after flecainide orally).

Conventional atrial strain parameters include strain curves with the reservoir phase, the conduit phase and contraction phase (*Badano L. P. et al. Eur Heart J Cardiovasc Imaging 2018;19:591-600*). In AF the conduit phase will continue until the end of ventricular diastole. Therefore, in patients with AF only two phases can be used to assess atrial strain; one during the reservoir phase (Sr) and a second one during conduit phase (Scd). However, in AF, Scd has the same value as LASr, but with a negative sign (*Badano L. P. et al. Eur Heart J Cardiovasc Imaging 2018;19:591-600*). A representative sample LASr measurement from one patient is shown on the Figure on the right.

These strain parameters will not be able to provide information about mechanical activation rates. In fact, looking carefully at sample strain curves from one pig in AF and with larger diastolic interval (*see Figure on the right*), the reviewer will notice the presence of small oscillations during the strain curve (yellow down arrowheads), which probably represent individual small contractions during AF. These oscillations are not considered on conventional strain analysis during a cardiac cycle. Moreover, at fast ventricular activation rates, these oscillations will be difficult to see. Strain analysis on 2D images at 60-80 Hz would also have limitations to identify the frequency of these oscillations compared to tissue doppler signals at >200 Hz for several beats (6-seconds segments in our study).

Based on this rationale about conventional strain parameters, we do not expect that they can provide the same information that the mechanical activation rates obtained with tissue Doppler signals. However, these parameters may provide additional information about “compliant” or “stiff” atria based on mean reservoir strain values (*Bax M. et al. Am J Cardiol. 2022;183:33-39*). In patients, we have measured left and right atrial strain reservoir (LASr and RASr, respectively) values using the four chamber-view of transthoracic echocardiography images, as recommended elsewhere (*Badano L. P. et al. Eur Heart J Cardiovasc Imaging 2018;19:591-600*). The results show that patients with successful pharmacological cardioversion have significantly higher LASr values than patients with non-successful pharmacological cardioversion. In the left atrium, we have also included the analysis of individual strain curves from different atria regions to calculate the dispersion of strain reservoir curves within the atrial wall. However, the latter did not show any significant difference between patients with successful and non-successful pharmacological cardioversion.

In the new version of the manuscript, we have included the information of LASr and RASr in the main table of the manuscript (*Table 1*). We have not included the analysis of regional strain reservoir dispersion since this analysis is not established in the literature and we have not seen differences. The reviewer can see below the results from this new analysis, although, only LASr and RASr have been included in Table 1.

Strain parameters on admission	Study population (N=83)	Pharmacological cardioversion (n=43)	Non-Pharmacological cardioversion (n=40)	p-value
LASr (%)	13.0±6.6	15.3±8.2	11.1±4.1	0.016
Dispersion of LASr (%)	34.2±49.6	38.5±66.1	30.6±30.8	0.966
RASr (%)	13.1±8.6	15.3±10.5	11.3±6.4	0.135
Dispersion of RASr (%)	33.9±51.3	28.7±40.8	38.5±59.4	0.376

We further tested whether a model based on LASr may be superior to the EMD model we had proposed. A LASr model showed an AUC 0.659 (sensitivity 53.8%, specificity 73.3%) to predict the primary outcome in the entire AF population (including paroxysmal and persistent AF patients). In the subset of persistent AF patients, the AUC decreased to 0.620 (Sensitivity 46.2%, specificity 75.9%). The EMD provided AUC 0.874 (sensitivity 79.3%, specificity 79.3%) to predict the primary outcome in the entire AF population. In the subset of persistent AF patients, the AUC was 0.875 (Sensitivity 84.6%,

specificity 81.5%). In the new version of the manuscript, we do not mention these comparisons, although we highlight in the statistical methods that we have selected the best predictive model.

4. Any data on TDI obtained from TEE vs TTE in AF patients?

In all patients with indication of transesophageal echocardiography (TEE), simultaneous TDI and surface lead II recordings were also obtained on TEE images from the left and right atria. In fact, EMD assessment on TEE sequences using the same formula as mentioned above (See the response to the first comment) also showed statistically significant differences between patients with successful and non-successful pharmacological cardioversion. See these data from Table 1 below:

Study variables on admission	Study population (N=83)	Pharmacological cardioversion (n=43)	Non-Pharmacological cardioversion (n=40)	p-value
Electromechanical dissociation (Hz)				
EMD in TTE	0.2±1.5	-0.7±1.2	1.2±1.2	<0.001
EMD in TEE	0.7±1.2	0.2±1.0	1.1±1.1	0.003

However, the number of patients with simultaneous TDI and surface lead II signals during TEE studies was 46, which probably limited the statistical power of this technique to provide a more robust model with higher or similar values of AUC than the model developed with transthoracic echocardiography (TTE). Importantly, the EMD model based on simultaneous acquisition of TDI signals and surface lead II recordings have the advantage of being non-invasive.

5. “Usefulness of tissue Doppler-derived atrial electromechanical delay for identifying patients with paroxysmal atrial fibrillation” by Kanako Akamatsu, et al. in Cardiovascular Ultrasound volume 18, Article number: 22 (2020) reported that the lateral EMD and the left atrial volume index were significant independent predictors for PAF patients. This paper reported a similar, though much more robust, data supporting the use of EMD as predictor of successful pharmacological cardioversion (AUC=0.874). This incremental knowledge should be expanded by other novel biomarkers and molecular mechanisms.

We thank the reviewer for giving us the opportunity to clarify the differences between the work by Akamatsu *et al.* (*Cardiovascular Ultrasound 2020*) and ours. The TDI-derived atrial electromechanical-delay (AEMD) described by the Akamatsu *et al.* measures the time from the beginning of the P-wave to the initial point of the spectral TDI-derived A´ on the septal and lateral sides of the mitral annulus. The authors retrospectively investigated whether TDI-derived AEMD would be useful to identify patients who had been diagnosed with paroxysmal AF in comparison with other variables known as strong predictors of AF, including LA volume index. In fact, they show that paroxysmal AF patients had longer AEMD, particularly for the lateral mitral annulus, compared with the healthy individuals and patients with multiple cardiovascular risk factors, although without AF. From our perspective, the AEMD parameter by Akamatsu *et al.* is far from what we propose in our study:

- First, we described using a fully translational and novel approach (from protein analysis, to the whole organ, large *in vivo* animals with AF and patients) to measure simultaneously atrial electrical and mechanical activity during AF. The latter is completely different to assessing atrial electromechanical activity during sinus rhythm, as in the work by Akamatsu *et al.* In fact, there are other options to assess atrial mechanical activity during sinus rhythm. Some of those are described in the Introduction of the manuscript. However, assessing atrial electromechanical activity during AF remains an unsolved problem and our study provides a novel approach to specifically address that.
- Second, our definition of electromechanical dissociation is based on faster electrical activation rates than the TDI-derived mechanical counterparts. Conversely, Akamatsu *et al.* described atrial

electromechanical delay as the time from the P wave to the spectral TDI-derived A', which does not indicate missing the mechanical activity after electrical activation.

- Third, our study further investigates the underlying mechanisms of atrial electromechanical remodeling using novel optical mapping techniques to simultaneously map transmembrane voltage and intracellular free calcium in whole heart preparations during short-lasting and persistent AF episodes. Moreover, this was done without using excitation-contraction uncouplers. Please, note that this approach is also highly novel and technologically challenging (see also Suppl. Figure 4) and have not previously reported in the literature. Further, proteomic studies and histopathology analysis were also done to demonstrate remodeling progression.
- And fourth, in the new version of the manuscript we further include biomarker and protein analyses, complemented with single cell computational simulations to demonstrate the main underlying mechanisms associated with electrical and mechanical dissociation.

We hope the reviewer can agree with us on the novelty and scientific value of our work compared to previous series.

As mentioned above, we have done a special effort to identify a biomarker that may reflect the underlying atrial damage during AF progression, which indeed could also be associated with contractile damage. Based on the fact that during AF the atria are activated at very rapid frequencies, this should be associated with high metabolic demands. Therefore, we tested whether AF could lead to cell death and atrial damage that could be detected using high-sensitivity Troponin T (hs-TnT) in plasma. In animals with AF (N=13) and sham-operated controls (n=6), we have analyzed hs-TnT in plasma samples at baseline, after 6 weeks of 100% AF burden (or the equivalent time-point in sham-operated controls) and at the end of follow-up in persistent AF (or the equivalent time-point in sham-operated controls). The equivalent times for comparisons in sham-operated controls were based on the average time to 6 weeks of 100% AF burden and the average time in persistent AF in AF animals. This analysis showed a progressive increase in hs-TnT levels in AF animals which was statistically significant during persistent AF compared to baseline values. This increase was not present in sham-operated controls. Moreover, at the end of follow-up hs-TnT levels were significantly higher in persistent AF animals than in sham-operated controls. These data have been included in a new Figure 5A and in the main manuscript (Page 7, lines 4-8). Although hs-TnT levels did not increase above the threshold for myocardial ischemia in clinical practice, these values are similar to those reported recently by Betül Toprak *et al.* (*Europace* 2023:Jan 4;euac260), suggesting the role of hs-TnI levels below conventional limits for predicting incident AF.

Although these results were encouraging, we further investigated whether this damage could be associated with the ventricles rather than the atria. We performed Caspase3 staining in left atrial and left ventricular samples of animals with AF at the end of follow up. These results showed a significantly higher number of Caspase3+ nuclei in the left atrium than in the ventricle of AF animals (new Figure 5B). Moreover, histopathology analysis of the left ventricular tissue in AF animals and sham-operated controls did not show any significant difference in fibrosis content (new Figure 5C), which further suggest that the increase in hs-TnT levels was associated with atrial damage. The lack of ventricular damage in AF animals was also confirmed with echocardiography studies during the follow up, which did not show any significant differences in left ventricular ejection fractions between AF animals and sham-operated controls (new Suppl. Figure 3). Additional analysis using immunostaining with TUNEL and confocal microscopy imaging also showed a higher number of TUNEL+ nuclei in the atria of AF animals compared to sham-operated controls (New Figure 5D). Moreover, Western blots of BAX in left atrial and ventricular samples of AF animals and sham-operated controls also supported that the increase in hs-TnT plasma levels in AF animals was associated with underlying AF induced-atrial damage (New Figure 5E). The reviewer can see the new Figure 5 below to facilitate the review process.

These results in pigs suggest that hs-TnT levels are able to detect low levels of continuous atrial damage in the atria of animals with AF. These results motivated further analysis of hs-TnT in plasma samples from patients using the same electrochemoluminometric assay in a Cobas e601 platform (Roche-Diagnostics, Basel, Switzerland). The results in patients did not show statistically significant differences between patients with successful and non-successful pharmacological cardioversion. We have included these new data in Table 1 as follows:

Study variables on admission	Study population (N=83)	Pharmacological cardioversion (n=43)	Non-Pharmacological cardioversion (n=40)	p-value
Hs Troponin T (pg/ml)	9.7±6.5	9.2±6.6	10.4±6.3	0.381

The results in patients may reflect specific conditions of the clinical scenario. Thus, all samples were taken with the patients in AF without a reference sample in sinus rhythm for comparisons with baseline values. Therefore, once AF is present, hs-TnT levels does not seem to be able to differentiate between different levels of remodeling in AF episodes lasting < 6 months (the limit for AF episodes in the study). In fact, in Figure 5A (see above), hs-TnT levels were not statistically different from early AF stages after 6 weeks of 100% AF burden and long-lasting AF episodes, which may explain the results observed in patients with all samples taken in AF.

6. Fig.3E showed that TDI signal significantly dropped below lead II atrial signal (suggesting EMD) starting from 9 weeks, while the proteomics data showed that proteins related to muscle contraction dropped starting from 6w. I believe that this implies that molecular biomarkers precede EMD. Further exploration of underlying mechanisms and molecular biomarkers will significantly increase enthusiasm. Confirmatory analysis by immunostaining and Western blots are also required.

The reviewer is right on the observation that TDI-derived activation rates significantly dropped below electrical activation rates signals from lead II at 9 weeks after the first recorded episode lasting ≥ 6 seconds (t_0) (Figure 3E, see also below). Proteomic data were obtained from right atrial biopsies at baseline (at the time of AV node ablation and beginning of the atrial pacing protocol to induce AF), at 6 weeks of 100% AF burden and at the end of follow up in persistent AF. Therefore, it is important to clarify that 6 weeks of 100% AF burden is not the same as 6 weeks of follow up. In fact, in Figure 3F we show that 6 weeks of 100% AF burden correspond to a mean follow up of 15 weeks (see also Figure 3E, F below).

However, we consider these comments relevant and we have performed additional confirmatory analysis to demonstrate the changes documented in proteomics. In fact, we have removed proteomics data from Figure 3 to simplify the understanding of Figure 3. We have included a new Figure 4 with proteomics and confirmatory analysis using histopathology, Real Time quantitative PCR, ELISA and Western blots. Overall, the confirmatory analyses support all AF-related remodeling changes observed in proteomics from atrial biopsies.

In brief proteomics analysis had revealed an increase in proteins associated with collagen-containing extracellular matrix and structural resistance of the extracellular matrix. This analysis also showed an early and significant decrease in proteins belonging to Gene Ontology categories associated with muscle contraction. These results were confirmed with an increase in interstitial tissue fibrosis as AF evolves (new Figure 4C, D), which was also associated with an increase in plasma levels of pro-collagen type III N-peptide (PIIINP) (New Figure 4E). The increase in fibrosis complexity was confirmed with an increase in gene expression levels of lysyl oxidase (new Figure 4F). Moreover, in the new version of the manuscript we have further explored the underlying damage in contractile proteins. More specifically, we performed Western blots of MYBPC3 and ACTC1, which showed a significant and early decrease (already after 6 weeks of 100% AF burden) of MYBPC3 in AF animals compared to baseline conditions (new Figure 4G, H). However, we could not document any significant decrease in ACTC1 (new Figure 4G, H). The new Figure 4 is also shown below to facilitate the review process. The new data are also in the main manuscript (Page 6, lines 23-33).

Figure 4. Fibrotic and contractile remodeling during atrial fibrillation progression in pigs. **A**, Time line of atrial biopsy studies. **B**, AF-related time-course changes of proteins belonging to muscle contraction, collagen-containing extracellular matrix (ECM) and ECM structural resistance. Color-coded circles represent the average of Z_q values of the proteins within each category, normalized to the baseline, that show a statistically significant change over the follow-up. A paired one-way-ANOVA test was used for comparisons. The rest of proteins with statistically significant changes over the follow-up are shown in Suppl Table 1. **C**, Representative examples of myocardial atrial biopsies (at 40X) from AF animals and sham-operated controls stained with Picrosirius red at different time points of follow up. **D**, **E**, **F**, Comparisons of fibrosis quantification (**D**), plasma pro-collagen type III N-peptide (PIIINP) (**E**) and gene expression levels of lysyl oxidase (**F**) between AF animals and sham-operated controls. **G**, Representative immunoblots of actin alpha cardiac muscle-1 (ACTC1) and the cardiac isoform of myosin-binding protein C (MYBPC3) in AF and sham-operated controls. **H**, Quantification and comparisons of ACTC1 and MYBPC3 expressions between AF animals and sham-operated controls. Immunoblot bands were quantified relative to total protein. *indicates equivalent time-points of the follow-up for sham-operated controls. 'B', 'I' and 'F', in panel G indicate Baseline, intermediate (i.e., 6 weeks of 100% AF burden or equivalent in controls) and final (i.e., persistent AF > 6 months or equivalent in controls).

Overall, the data support that atrial muscle contraction is affected at early stages of AF progression, which is associated with other structural changes in atrial fibrosis and complexity of fibrosis content. Moreover, the analysis of lead II atrial tracings also showed an increase in atrial activation rates, which is compatible with underlying electrical remodeling, as reported elsewhere in more detail (*Raphael P. Martins et al. Circulation. 2014;129:1472-1482*). Additional changes in calcium handling proteins may significantly contribute to electromechanical dissociation. Importantly, the advantage of the animal model is that provides an experimental scenario without additional comorbidities and confounding factors, commonly present in samples from patients undergoing cardiac surgery. Moreover, in the animal model is possible to know the entire AF history and time in AF, which are also challenging in patients.

To further complement the comments indicated by this reviewer, we have performed Western blots of phospholamban (PLN) and Thr17 phospho-PLN, Ryanodine receptor 2 (RyR2) and Ser2814 phospho-RyR2, sodium/calcium exchanger (NCX), and sarco-endoplasmic reticulum ATPase2 (SERCA2). The analyses were done in right atrial samples from long-lasting persistent AF animals and sham-operated controls at the end of follow-up. The main results are shown in a new Suppl. Figure 7. In brief, the main changes were observed in PLN and Thr17pPLN with a significant decrease in the ratio

Thr17pPLN/Total-PLN in AF animals compared to Sham-operated controls (See new Suppl. Figure 7A, below).

Total RyR2 and Ser2814pRyR2 were also significantly reduced in AF animals compared to Sham-operated controls. However, the ratio Ser2814pRyR2/ Total-RyR2 did not show statistically significant differences between groups. The remaining comparisons with NCX and SERCA2 also showed no statistically significant changes between animals with AF and sham-operated controls (Suppl. Figure 7). The overall results are also described in the main manuscript (Page 8, lines 23-33).

Finally, we performed single cells simulations to understand the effect of electrical and calcium handling remodeling on voltage and calcium dynamics compared to baseline modeling conditions without remodeling. Human atrial action potential and its corresponding calcium transients were simulated based on the mathematical model described by Skibsbbye *et al.* (*J Mol Cell Cardiol.* 2016;101:26-34). We specifically tested 3 scenarios using irregular pacing to simulate the irregular rhythm of AF: *i*) AF without remodeling (based on the model by Skibsbbye *et al.*), *ii*) AF with electrical remodeling (based on ion current changes of AF remodeling in *J Mol Cell Cardiol.* 2016;101:26-34), and *iii*) AF with electrical remodeling and additional decrease of -82% of Thr17 phosphorylation of phospholamban/Total phospholamban, which was adjusted according to our experimental results. Please, see also below the *panel A of the new Suppl. Figure 8*, in which we specifically describe the remodeling changes incorporated in modeling scenarios *ii*) and *iii*).

The simulations were performed using random irregular pacing rates from 3 to 7 Hz for 6 seconds after a steady state of 4-second pacing at 7 Hz. The results showed significantly higher values of voltage and calcium dissociation in the models with atrial remodeling (scenario (ii): electrical remodeling and scenario (iii): electrical + calcium handling remodeling) compared to simulations in baseline conditions without any remodeling. These data support our experimental results *ex vivo* in isolated heart preparations, and *in vivo* during simultaneous electrical and TDI recordings in the animal model and patients. More specifically, these experimental results showed consistently that electromechanical dissociation was statistically associated with the presence of atrial remodeling (Figure 2, 6, 7). In other words, the presence of electromechanical dissociation during AF is not only a frequency-dependent effect. Similar to the experimental results, in the computational simulations we also see that simulations without remodeling showed lower values of voltage and calcium dissociation than the same simulations with remodeling. These data have been included in a new Suppl. Figure 8. The results have been also included in the main manuscript (Page 9, lines 1-9). The main results in Suppl. Figure 8D are also attached on the right.

7. Overall, the manuscript is very hard to read and will need re-writing in many parts.

We have re-written substantial parts of the manuscript aiming to improve the understanding of the entire study. All changes compared to the previous version of the Manuscript and Suppl. Material are highlighted in red. The text has been written in a more concise manner to include all the new data and improve the understanding. We have also moved specific details of the Methods to the Suppl. Material to keep in the main manuscript the most essential parts. We have simplified some of the main Figures, although this required increasing the number of figures to enable us including the new analyses and results requested by the reviewers.

Minor points:

1. Line 131: average of 376.4 +/- 111.7 days and 702.9 +/- 189.2 days show a huge variability in the timing. This needs justification and explanation.

The variability of the follow up in the animals with long-lasting persistent AF is explained by the intrinsic animal-specific variability to develop persistent AF, which was defined as in the clinic (at least ≥ 7 days in self-sustained persistent AF). In some animals persistent AF can be developed for example in 12 weeks and in others may take longer. However, we did not consider that persistent AF started until the animal fulfilled the criterium of ≥ 7 days of self-sustained persistent AF.

The average time in persistent AF also showed variability because we wanted assess the electromechanical relationship at different intervals in persistent AF. Please, be aware that, first we studied the atrial electromechanical relationship in animals with healthy atria and we did not observe dissociation (Figure 1). Then we tested the atrial electromechanical relationship in persistent AF and we could not know whether the dissociation would be present early during AF remodeling or it might have needed several months in AF. That was the reason to include animals with different times in persistent AF in Figure 2. The results in persistent AF animals with different follow up periods showed overt electromechanical dissociation. Therefore, the next step was to determine the time line of electromechanical remodeling progression. The latter required a new series of animals (Figure 3) in which we demonstrated that electromechanical remodeling is an early parameter of remodeling progression. Overall, the total number of animals included in the study for *in vivo* and *ex vivo*

experiments was 52. The latter required a multidisciplinary team of investigators and a lot of resources to manage these large mammals and all the data generated.

2. Fig. 4 is very confusing; Any data of AF without remodeling in Fig.4E?

Figure 4 (Now Figure 6 in the new version of the manuscript) includes optical mapping data of transmembrane voltage and intracellular free calcium of isolated hearts from animals with persistent AF (median time in self-sustained persistent AF 10.0 months) and control hearts from healthy animals. The mapping data were obtained from short-lasting AF episodes after burst pacing (in controls) and spontaneous self-sustained AF episodes in the hearts from persistent AF animals. Moreover, in 9 of 13 hearts, we also performed voltage and calcium recordings after the administration of Acetylcholine (ACh) into the perfusate to address whether the presence of voltage and calcium dissociation was only a frequency-dependent effect or it was also related to atrial remodeling. In the new version of the manuscript, we have included new data from 2 hearts from animals with persistent AF to reply to one specific comment from other reviewers.

Therefore, the new results include data from AF episodes in hearts with remodeling and without remodeling. See below panel E, F from new Figure 6 in which the reviewer can see the labels related to each experimental setting (color-filled triangles), the individual results for each optical recording included in the analysis (Panel E) and the quantification of the results (Panel F). The reviewer will be able to see that the results support that voltage and calcium dissociation during AF is not only a frequency-dependent phenomenon, but it is also due to the underlying AF-related atrial remodeling (electrical and calcium remodeling). In fact, during AF movies with activation frequencies <6.5 Hz, there was a significantly higher voltage and calcium dissociation in hearts from persistent AF animals than in controls. This is also consistent with the new results showed in single cell simulations in Suppl. Figure 8.

Reviewer #3 (Remarks to the Author):

Manuscript Titled, Non-invasive electromechanical assessment during atrial fibrillation identifies underlying atrial myopathy alterations with early prognostic value, by Enríquez-Vázquez, etc. presents a critical parameter, Atrial electromechanical dissociation (EMD) in the prognostic prediction of atrial fibrillation (AF) development. The study developed a method of EMD calculation and evaluated it in animals (preclinical) with and without AF and even in the clinical cohort, which showed interesting results and significance. The best part of the study is that it suggested EMD is earlier than biomarkers in atrial remodeling and AF prediction, I have a few comments:

We thank the reviewer for his/her main comment about the relevance of electromechanical dissociation as an early remodeling parameter during AF progression. In fact, we also think this is one of the main messages of the manuscript.

- Regarding the DF of EMD, how to explain that, in paroxysmal AF, DF is -0.61Hz?

This is a relevant question. There is a simple explanation for it; electromechanical assessment using simultaneous lead II-derived atrial ECG signals and TDI signals combines local information of the TDI signals with more general atrial information from the lead II surface ECG. Therefore, in healthy atria and specific atrial sites, the mechanical activation rates could be higher than the electrical activation rate detected on the lead II surface ECG, which explains a negative value. This situation was not present during electroanatomical mapping procedures in persistent AF animals, in which we specifically positioned a multipolar catheter at the same atrial region where TDI signals were acquired from (Figure 2B, C, E). Importantly, in such acquisitions in persistent AF animals we also had the information from lead II surface ECG, so we could also analyze if EMD assessment using the surface ECG and TDI signals could have significant limitations to assess EMD. The reviewer can see in Figure 2D, F (See also below) that both approaches, either using simultaneous acquisition of surface lead II ECG signals, or spatially colocalized intracardiac electrograms, with TDI signals will be able to reliably assess EMD. The reviewer will also see in Figure 2F (below) the difference between persistent AF animals (orange-filled triangles) and healthy controls (green-filled triangles) using the same approach for electromechanical assessment with simultaneous acquisition of lead II surface ECG signals and TDI signals. The presence of EMD was negligible in the animals with healthy atria without AF-related remodeling.

To further address why lead II surface ECG is sufficient to assess electromechanical activity, in persistent AF animals we also performed full electroanatomical mapping of both atria at multiple locations. These data enabled us to calculate electrical activation rates (using DF) at multiple locations and at the same time having the surface ECG from lead II during the mapping procedure. Interestingly, $70.9 \pm 11.8\%$ of the atrial endocardial surface showed local DF values within the 5th-95th percentiles of lead II-derived atrial electrical activation rates during the mapping procedure. Therefore, most atrial regions will show atrial activating rates that can be detected in dominant frequency values of lead II-derived atrial ECG signals. This analysis is mentioned in Page 5, lines 17-20 of the main manuscript

and in Suppl. Figure 2. Please see below Panel B, C of Suppl. Figure 2 to facilitate the revision of the main message.

Moreover, in patients, electromechanical assessment included information from the left and right atria to have a more complete assessment of the electromechanical activity in the atria. EMD was calculated as follows:

$$EMD = \frac{EMD_{right\ atrium\ (RA)} + EMD_{left\ atrium\ (LA)}}{2} = \frac{(EAR - MAR_{RA}) + (EAR - MAR_{LA})}{2}$$

Where, EAR: electrical activation rates; MAR: mechanical activation rates; RA: right atrium; LA: left atrium.

This approach decreases the probability of any potential bias from TDI-derived mechanical activation rates that will not be representative for comparisons with the electrical atrial frequencies derived from lead II.

Therefore, it is possible to see faster mechanical activation rates than their mechanical ECG-derived counterparts in healthy atria. This will happen if the TDI data comes from areas with fast electrical activation rates (e.g., the left atrial appendage in panel C above, with the magenta color indicating and electrical activation frequency at 9.6 Hz) and their mechanical counterparts are also able to follow such activation frequencies. In such scenario, lead II-derived atrial activation rates may show slower electrical activation rates than TDI-derived mechanical activation rates and the EMD values will be <0. In any case, this will indicate a healthy atrium and lack of EMD. Moreover, the opposite, having slow TDI-mechanical activation rates from an atrial region with very low electrical activation rates, which may be not representative for the rest of the atria, will be very unlikely based on the mechanical activation frequencies we observed once dissociation is established (3.5-5 Hz, in Figure 3E). In patients, this effect from one specific region will be also minimized since electromechanical assessment included information from the left and right atria. It is also important to mention that TDI information was obtained from the same atrial regions among patients and animals, which makes the data consistent for comparisons.

In the new manuscript, we also discuss briefly this potential concern in Page 13, lines 28-32 and Page 14, lines 1, 2.

- If the EMD could be done at the segmentation level, the distribution of EMD could provide tips for AF ablation. Is there any difference in the atrial distribution of EMD?

This is a great comment. In fact, the ideal scenario would be to have TDI-derived mechanical information from the entire atria to be able to assess mechanical activity from different regions. From a practical perspective in patients with AF, this technology would require 3D tissue Doppler imaging at high acquisition rates (>200 Hz). To our knowledge, this is not available and it would be highly computational demanding on current echocardiography machines. Hopefully, our work will motivate other investigators and companies to achieve that goal in the near future.

Nowadays, the realistic scenario is to acquire 2D TDI information from different atrial regions as we have done in patients (from the left and right atria). Despite regional analysis can be focused on specific atrial regions of the atria, having information from more than one region may provide a more complete assessment of the underlying atrial myopathy. Our approach was to integrate the information from both atria to assess EMD. Based on this comment, we specifically analyzed electromechanical data from the right and left atria independently. Interestingly the right atrium seems to be more sensitive to electromechanical dissociation at electrical activation rates <6.5 Hz. Please, see this analysis on the right. However, we have not included this subanalysis in the main manuscript since the overall assessment (including the left and right atria) provided the best predictive model for the primary outcome and AF recurrences at 1 year of follow-up (AUC 0.875).

Electromechanical assessment in the pig model included TDI data from the posterior left atrium. The data was obtained with transesophageal echocardiography images, since the transthoracic window in the pigs did not enable us an optimal apical four chamber view with appropriate tissue Doppler beam alignment. In the new version of the manuscript, we have included new animals (n=7) with healthy atria to obtain TDI data from the posterior left atrium and the left atrial free wall during short-lasting AF episodes induced by burst pacing. In persistent AF animals, we reviewed the previous cases with invasive mapping and in 7 of 10 animals we had TDI data with spatially colocalized and simultaneous intracardiac electrograms from the posterior left atrium and the left atrial free wall. We have also included one more persistent AF animal.

Overall, these new data show that in pigs with healthy atria, there are no signs of electromechanical dissociation either using simultaneous lead II ECG signals with TDI data from the posterior left atrium or the left atrial free wall. These new results have been included in the new version of Figure 1 (panels G, H. See also below) and in the main manuscript (Page 4, lines 29-32).

The number of animals and movies used for analysis have been also updated in Figure 1F (N=16, which includes the previous 9 animals with only data from the posterior left atrium and new 7 animals, with data from the posterior left atrium and left atrial free wall).

Conversely, in persistent AF animals, the analysis of spatially synchronized simultaneous TDI signals and intracardiac electrograms showed consistently the presence of electromechanical dissociation in all movies either from the posterior left atrium or the left atrial free wall. This new analysis is shown in Figure 2E (see also below) and in the main manuscript (Page 5, lines 14-16). The number of persistent animals and movies used for analysis have been also updated in Figure 2D, F.

Overall, regional analysis of EMD seems to support that atrial mechanical activity may be affected quite homogeneously among regions, at least in the left atrium. In pigs, we did not have data from the right atrium since right atrial views using transesophageal echocardiography in pigs are quite challenging. However, we would expect also electromechanical dissociation on the right atrium since human data showed that this atrium could be more sensitive to dissociation at lower atrial activation frequencies.

- The study showed atrial area increased during atrial remodeling as usual. How the size of the area affects the EMD value? As we know, in clinical scenarios, some AF patients have average atrial size.

In pigs, we have sequential atrial size measurements and electromechanical data at 3-week intervals during the follow-up (Figure 3B and E, respectively). These data have enabled us to study the association of time-course changes in left atrial size (indexed to weight) with the onset of electromechanical dissociation. We specifically selected the data from 3 specific time points during the follow-up: Baseline, at the time of electromechanical dissociation and at the end of follow-up. The analysis was done by a multilevel mixed-effects linear regression with repeated measures. Interestingly, we did not observe any statistically significant association between left atrial size in the pigs and the onset of electromechanical dissociation ($p=0.42$). This further supports that electromechanical dissociation is an early parameter of AF-related atrial remodeling, before overt relevant changes in atrial size.

In patients, we do not have sequential electromechanical data during the follow-up. However, we performed a correlation analysis between left atrial size and electromechanical dissociation. This analysis did not show any statistically significant correlation between the left atrial size and the presence of electromechanical dissociation ($p=0.43$, $Rho=0.104$). These results are consistent with the inclusion criteria, aiming to include patients at early remodeling stages of AF progression (episode duration ≤ 6 months based on symptoms) and without other relevant comorbidities. In fact, the study population was composed of relatively young patients (55.0 ± 11.5 years old) with low rates of comorbidities (Table 1 and Figure 7B) and average 3D left atrial volume index values (32.6 ± 8.6 ml/m²) below the dilation threshold (34 ml/m²).

In the manuscript, we highlight that “*the clinical results further supported the relevance of EMD as an early indicator of atrial remodeling before other changes in conventional echocardiography parameters or blood biomarkers (Table 1) became relevant*” (Page 11, lines 23-25).

- How to explain, in PsAF remodeling heart, the voltage-calcium dissociation is not available in high frequency.

Optical mapping data in isolated heart preparations were obtained from short-lasting AF episodes after

burst pacing (in controls) and spontaneous self-sustained AF episodes in the hearts from persistent AF animals. In the hearts from persistent AF animals, we already had documented voltage and calcium dissociation without administering Acetylcholine (ACh) into the perfusate. Therefore, we did not need to further increase the activation frequency to see voltage and calcium dissociation. However, in the healthy atria of control animals, voltage and calcium dissociation was negligible at activation frequencies <6.5 Hz. Therefore, in controls we also included recordings with ACh into the perfusate to increase the activation frequency. This demonstrated that in the healthy atria, voltage and calcium dissociation will also be present at high activation rates, specially >8.5 Hz.

To address this concern, we have included 2 more optical mapping studies in hearts from animals with persistent AF. In these hearts, after the acquisition of simultaneous voltage and calcium recordings during spontaneous AF, we also acquired voltage and calcium data with ACh into the perfusate. A total of 45 new optical recordings have been included in the analysis. As expected, voltage and calcium dissociation increased further as the activation frequency was higher with ACh into the perfusate. The individual results for each optical movie included in the analysis and the quantification of the results are shown in the new Figure 6E, F (See these 2 panels below). The reviewer will see that the results support that voltage and calcium dissociation during AF is not only a frequency-dependent phenomenon, but it is also due to the underlying AF-related atrial remodeling (electrical and calcium remodeling). In fact, during AF movies with activation frequencies <6.5 Hz, there was a significantly higher voltage and calcium dissociation in hearts from persistent AF animals than in controls. A mixed model based on Generalized Estimating Equations (GEE) was used to test for significant differences and interactions. This GEE model prevents any potential bias from the hearts with more optical recordings for analysis than the rest.

Reviewer #4 (Remarks to the Author):

This is an interesting and very comprehensive study that addresses several aspects of electromechanical characterization during atrial fibrillation (AF). This study reports novel mechanistic insights into the electromechanical remodeling process associated with AF progression and translates its prognostic value in the clinic. The study provides experimental support from a pig model in which sequential electromechanical assessment during AF progression showed a progressive decrease in mechanical activity and early dissociation from its electrical counterpart during in vivo studies. Novel optical mapping techniques applied to the contracting atria of Langendorff-perfused hearts during AF demonstrated underlying dissociation of transmembrane voltage and intracellular calcium transients supporting the in vivo results. Atrial biopsies at different stages of AF-related atrial myopathy further revealed early downregulation of contractile proteins and progressive increase in biological functions associated with atrial fibrosis. In patients, non-invasive assessment of the atrial electromechanical relationship during AF enabled novel characterization of the underlying atrial myopathy at early stages of AF progression. Atrial electromechanical dissociation was a prognostic indicator for acute pharmacological cardioversion, and also performed better than other conventional cardiac imaging parameters to early predict AF recurrences during the follow-up.

This study provides several significant novel findings that include identifying electromechanical dissociation as an early marker of atrial remodelling progression during AF, novel optical mapping ion channel dynamics and validated TDI in a clinical setting.

We thank the reviewer for the comments about the substantial novelty of our work and its value to provide novel insights into AF pathophysiology.

I have minor comments on this excellent manuscript.

1) Electromechanical assessment in the pig model: Please provide more details on how you defined short lasting AF in this model i.e. < 5 minutes, etc.

In vivo short-lasting AF episodes in control animals with healthy atria were induced with burst pacing at 10 Hz. The median duration of such episodes was 23.5 seconds [interquartile range: 11.5, 56.0 seconds].

In Langendorff-perfused isolated heart preparations, AF episodes in control hearts were also short-lasting after burst pacing at 10 Hz. The median duration of these episodes was 18 seconds [interquartile range: 11.0, 37.0 seconds]. The administration of acetylcholine into the perfusate increased fibrillatory frequency (Figure 6E) and made the episodes long-lasting without spontaneous termination in control hearts.

We have included the *in vivo* quantification of short-lasting AF episodes duration in the main manuscript (Page 4, lines 24-26). None of short-lasting AF episodes in healthy animals lasted more than 3 minutes (without Acetylcholine). Therefore, we also included the 3-minute limit to defined short-lasting AF episodes (Page 15, line 2).

2) Was the threshold to achieve EAR steady state affected by burden of AF preceding this finding?

In all animals, AF burden was at 100% by the time electrical activation rates reached the steady state. In other words, 100% AF burden preceded the plateau of electrical activation rates which we documented between 21-27 weeks after protocol initiation (Page 6, lines 7-10). Interestingly, electromechanical dissociation was also commonly documented before 100% AF burden in the AF animals with sequential 3-week follow-up intervals. More specifically, electromechanical dissociation was documented at 89.1±19.9 % AF burden in our pig model of persistent AF. Please, note that AF burden 100% does not mean that AF episodes will be persistent, since these episodes can still finish

before 7 days. In fact, at the time of 100% AF burden is reached, the average AF episode duration was 1.17 ± 1.47 hours. However, upon another burst of 30-second rapid pacemaker stimulation, AF is induced and the next episodes could last again several hours or days. Therefore, 100% AF burden represents a situation where the time in sinus rhythm is minimal during the 24-h day; after the pacemaker senses sinus rhythm for 6 seconds, the pacemaker immediately activates the rapid pacing protocol in the right atrium to induce AF. In any case, we only defined persistent AF when episodes were self-sustaining for ≥ 7 days.

In the new version of the manuscript, we specifically mention that electromechanical dissociation was documented early during AF progression at the time of 89.1 ± 19.9 % AF burden (*Page 6, lines 10-11*).

Interestingly, rather than AF burden, the most relevant parameter associated with the time to the steady state of electrical activation rates is the slope of electrical activation rates changes over time (dDF/dt). Thus, animals with rapid acceleration of electrical activation rates will reach such a steady state faster than other animals with lower acceleration in the activation frequency. Although the total number of animals in the study was very large ($N=52$) to complete all experimental questions, the number of animals with sequential monitoring at 3-week intervals was limited (13 AF animals and 6 sham-operated controls) to reach robust conclusions for this comment. However, our assertion to address this comment is based on a larger number of AF animals ($N=31$) from an ongoing study. We are glad to share these analyses with the reviewer. See on the right, one Figure that supports the explanations in this paragraph. The Figure shows natural log plots of the slope of EAR change (calculated with the dominant frequency [DF] of atrial electrograms) versus time to completion of AER in the pig model of AF.

3) Was there a direct correlation between LA area and EARs?

In pigs, we took advantage of the sequential follow-up at 3-week intervals to analyze if there is any correlation between LA size (LA diameter indexed to body weight) and electrical activation rates (EARs). Using a multilevel mixed-effects linear regression with repeated measures, we did not identify any significant correlation between LA size and electrical activation rates ($p=0.70$).

In patients, we performed a Pearson's correlation coefficient test to analyze the relationship between LA area and electrical activation rates in our human cohort. We found no association between the two variables ($r=0.108$, $p=0.400$). We repeated the analysis using the LA volume index and the results were similar ($r=0.083$, $p=0.529$).

Therefore, LA size does not show a significant association with EARs. Other functional factors must be involved.

4) Can serial pharmacological cardioversions prevent reaching the plateau? In other words, can interventions prevent AF from progressing to persistent or complete electrical remodeling?

This is an interesting question with clinical implications to potentially stop remodeling progression. We did not cardiovert our animals during the follow-up to specifically address this question. Therefore, using the data from the animals included in the study we could not answer this question. However, over the last 6 years working with the pig model of AF, in 3 animals we have seen an increase in the threshold to capture the right atrium that interrupted atrial pacing during the 3-week interval between

interrogations. Therefore, in those animals the AF burden dropped to 0% at the time of the next interrogation. Once the output of the atrial lead was adjusted, we could reinduce AF again. Therefore, in these 3 cases we had the opportunity to analyze the electrical activation rates at: 100% AF burden before the burden decreased to 0%, upon the atrial lead output was adjusted and we recorded the new first AF episode, and after AF burden increased again to 100%. The 3 sample cases are shown in the Figure on the right. The reviewer can see that dominant frequency (DF) values decreased significantly in all cases during the first episodes after adjusting the atrial lead output for capture. However, this remodeling rapidly progressed within the next follow-up interval (3 weeks) to the same levels of DF values that the animals had before losing atrial capture.

These data indicate that atrial electrical remodeling could be reversed during early stages of AF progression. However, this might not apply to persistent AF, especially after long periods in AF. A hallmark of remodeling in the persistently fibrillating atria is the presence of a constitutively active, parasympathetic stimulation independent, acetylcholine (ACh) sensitive inward rectifier K⁺ current (I_{KACH}) (Makary S et al. *Circ Res.* 2011;109:1031-43. Voigt N et al. *Circ Arrhythm Electrophysiol.* 2010;3:472-80). The latter might prevent reverse electrical remodeling after cardioversion. Moreover, specific myocardial diseases may also affect electrical remodeling and prevent this reverse electrical remodeling after cardioversion. We briefly discuss the role of patient-specific mechanical remodeling in Page 12, lines 14-18.

5) There was 1 pig that never achieved the steady state what protected this pig, there may be some interesting insights derived from this pig hat may provide information on a protective innate mechanism that allows some to never progress to persistent AF. I understand this is only 1 animal but important information may be obtained from this finding.

We agree that this animal is very interesting to further investigate animal-specific characteristics that may explain a much slower atrial electromechanical remodeling progression. Despite each animal showed individual-specific time-course changes in atrial remodeling, this one was particularly slow. As we show in the Figure included in the response to comment#2 of this Reviewer, there are animal-specific slopes of electrical remodeling progression. In fact, we recently reported that atrial electrical remodeling also follows patient-specific patterns in the clinic (Lillo-Castellano et al. *Europace* 2020;22:704-715). This also applies to electromechanical remodeling, although in the overall series TDI-derived mechanical activation rates significantly dropped below lead II-derived electrical activation rates at 9 weeks after the first recorded episode lasting ≥ 6 seconds (t0).

In the specific animal with particularly slow progression of remodeling without electromechanical dissociation at the end of the follow-up (48 weeks after t0) we did not find any prominent parameters that could distinguish this animal from the rest. At the end of the follow-up, when all animals showed electromechanical dissociation, the left atrial diameter was 4.35 ± 0.53 cm (indexed to body weight) compared to 4.50 cm in the animal with slow progression. We also compared atrial electrical activation rates between this animal and the rest. Dominant frequency values at the end of follow-up in this animal were 6.23 Hz, which were slightly lower than the mean for the other animals (7.35 ± 0.95 Hz), although still higher than the minimum frequency in another animal showing electromechanical dissociation at the same follow-up (5.98 Hz). Therefore, other underlying mechanisms must be involved, some of which may be related to the new experimental data and single cell simulations presented in the new Figures 4, 5 and Suppl. Figures 7, 8.

6) Calcium transient alternans: Can you indicate what % of the hearts showed dissociation that was preceded by calcium transient alternates in the text sentence 229.

In 5 of 7 (71%) hearts from controls and in 4 of 6 (67%) hearts from animals with persistent AF, voltage and calcium dissociation was preceded by calcium transient alternans. This information has been added to the new version of the manuscript (Page 8, lines 19-22). A sample case of calcium alternans is shown in the new Suppl. Figure 6A.

7) Can you provide information whether cardioversions were performed with or without antiarrhythmic drug therapy i.e. amiodarone?

None of patients were taking antiarrhythmic drugs before the inclusion in the study and at the time of electromechanical assessment. This is an important comment, since antiarrhythmic drugs might have potentially affected electromechanical assessment. In the results section we had mentioned that “Patients under baseline treatment with antiarrhythmic drugs (including calcium channel blockers and digoxin) other than β -blockers were excluded”. In the new version of the manuscript, we have included more specific details in the Methods to address this comment: “More specifically, patients taking amiodarone within the 6-month period before inclusion, or other antiarrhythmic drugs within the month before inclusion, were excluded” (Page 16, line 32 and Page 17, lines 1-2).

8) Acute cardio version is an important endpoint, however any data on long-term maintenance of sinus rhythm?

The primary outcome was cardioversion to sinus rhythm within a 24-h window after flecainide administration orally. The study secondary outcomes were pharmacological cardioversion of persistent AF and AF recurrences at 1-year follow-up.

Regarding the specific question of AF recurrences, Kaplan-Meier curves and log-rank test analysis showed a higher AF recurrence rate at 1-year follow-up ($p=0.002$) and 2.5-year follow-up ($p=0.003$) in patients with $EMD \geq 0.32$ Hz (new Figure 9D). In contrast, atrial dilation using 3D left atrial volume index or right atrial area criteria (34 ml/m^2 or $11 \text{ cm}^2/\text{m}^2$) did not differentiate between patients with or without AF recurrences during the follow-up (new Figure 9E, F). Similar results were obtained in the persistent AF patient subset, in which EMD, but not left or right atrial dilation criteria, detected baseline remodeling changes with prognosis impact on recurrences (new Figure 9 G-I).

Altogether, these results support the relevance of EMD as an early indicator of atrial remodeling before other changes in conventional echocardiography parameters or blood biomarkers (Table 1) become relevant.

The text with these results is in Page 11, lines 12-17.

9) Discussion: It may be possible that the biomarkers measured did not provide early evidence or predicted EMD, i.e. BNP. Newer biomarkers may be more relevant.

As mentioned in our response to Comment#5 from Reviewer#2, we have done a special effort to identify a biomarker that may reflect the underlying atrial damage during AF progression, which indeed could also be associated with the decrease in proteins associated with muscle contraction we documented in proteomics analysis (Figure 4A, B). Based on the fact that during AF the atria are activated at very rapid frequencies, this should be associated with high metabolic demands. Therefore, we tested whether AF could lead to cell death and atrial damage that could be detected using high-sensitivity TroponinT (hs-TnT) in plasma. In animals with AF ($N=13$) and sham-operated controls ($n=6$), we have analyzed hs-TnT in plasma samples at baseline, after 6 weeks of 100% AF burden (or the equivalent time-point in sham-operated controls) and at the end of follow-up in persistent AF (or the equivalent time-point in sham-operated controls). The equivalent times for comparisons in sham-operated controls were based on the average time to 6 weeks of 100% AF burden in AF animals, and the average time in persistent

AF in AF animals. This analysis showed a progressive increase in hs-TnT levels in AF animals which was statistically significant during persistent AF compared to baseline values. This increase was not present in sham-operated controls. Moreover, hs-TnT levels were significantly higher in AF animals than controls at the end of follow-up in persistent AF. These data have been included in a new Figure 5A and in the main manuscript (Page 7, lines 4-8). Although hs-TnT levels did not increase above the threshold for myocardial ischemia in clinical practice, these values are similar to those reported recently by Betül Toprak *et al.* (*Europace* 2023:Jan 4;euac260), suggesting the role of hs-TnI levels below conventional limits for predicting incident AF. In pigs, we further performed additional Western blots and immunostaining analyses to demonstrate that the increase in plasma levels of hs-TnT was related to atrial damage (Figure 5).

These results in pigs suggest that hs-TnT levels are able to detect low levels of continuous atrial damage in the atria of animals with AF. These results motivated further analysis of hs-TnT in plasma samples from patients using the same electrochemoluminometric assay in a Cobas e601 platform (Roche-Diagnostics, Basel, Switzerland). However, the results in patients did not show statistically significant differences between patients with successful and non-successful pharmacological cardioversion. We have included these new data in Table 1 as follows:

Study variables on admission	Study population (N=83)	Pharmacological cardioversion (n=43)	Non-Pharmacological cardioversion (n=40)	p-value
Hs-TroponinT(pg/ml)	9.7±6.5	9.2±6.6	10.4±6.3	0.381

The results in patients may reflect specific conditions of the clinical situation that are different from the study in pigs. Thus, in patients all samples were taken in AF without a reference sample in sinus rhythm for comparisons with baseline values. Therefore, once AF is present, plasma levels of hs-TnT does not seem to be able to differentiate between different levels of remodeling in AF episodes lasting ≤ 6 months (the limit for AF episodes in the clinical series). In fact, in Figure 5A (see also on the right. The asterisks indicate that samples in sham-operated controls were taken at equivalent time points of follow up compared to AF animals), hs-TnT levels were not statistically different from early AF stages after 6 weeks of 100% AF burden and persistent AF stages. The latter may explain the results observed in patients with all samples taken in AF.

REVIEWERS' COMMENTS

Reviewer #1 (Remarks to the Author):

The authors have addressed my questions satisfactorily. I have no further comments. However, I would like to know why 4 authors were added during this revision. Did they contribute any new data? Please specify.

Reviewer #2 (Remarks to the Author):

The authors have addressed my concerns.

Reviewer #3 (Remarks to the Author):

The authors thankfully responded to my comments in detail and updated the manuscript to reflect these changes.

Reviewer #4 (Remarks to the Author):

This is a revised version of a manuscript that reports novel mechanistic insights into the electromechanical remodeling process associated with AF progression, this is a translational study that provides further support to the experimental findings in the clinical setting. In pigs, sequential electromechanical assessment during AF progression showed a progressive decrease in mechanical activity and early dissociation from its electrical counterpart. Atrial tissue samples from animals with AF revealed an abnormal increase in cardiomyocytes death and alterations in calcium handling proteins. High-throughput quantitative proteomics and immunoblotting analyses at different stages of AF progression identified downregulation of contractile proteins and progressive increase in atrial fibrosis. Moreover, novel optical mapping techniques, applied to whole heart preparations during AF, demonstrated that AF-related remodeling decreases the frequency threshold for dissociation between transmembrane voltage signals and intracellular calcium transients compared to healthy controls. These results were also confirmed in single cell simulations of human atrial cardiomyocytes. In patients, non-invasive assessment of the atrial electromechanical relationship further demonstrated that atrial electromechanical dissociation was an early prognostic indicator for acute and long-term rhythm control.

This is a very comprehensive translational study that provides novel insight into the mechanisms that lead to progression from paroxysmal to persistent AF. The findings suggest that early electromechanical dissociation seems to precede all other structural and ultrastructural changes that may explain the progression of AF.

The investigators have extensively revised the manuscript and have provided detailed responses to all the comments of the 4 reviewers. Additionally, the investigators have performed additional experiments to further provide support for the initial findings.

I have no additional comments.

Reviewers' comments

Reviewer #1 (Remarks to the Author): The authors have addressed my questions satisfactorily. I have no further comments. However, I would like to know why 4 authors were added during this revision. Did they contribute any new data? Please specify.

The final version of the manuscript required a large number of additional experiments and new analyses to address all questions. The 4 new authors we had included in the previous version of the manuscript significantly contributed to such experiments, data analyses and manuscript revision. More specifically:

- Marinela Couselo-Seijas has been in charge of all Western blots analyses in tissue samples from AF animals and sham-operated controls. She also performed new histopathology and immunohistochemistry analyses to address additional questions from the reviewers.
- Álvaro García-Osuna and Jordi Ordonez-Llanos were in charge of the analysis related to blood biomarkers. More specifically, all hs-cTnT analyses from animal and patient samples were done by both of them at the Institut de Recerca de l'Hospital Santa Creu i Sant Pau (Barcelona, Spain). Both co-authors are active investigators on the topic of biomarkers related to cardiac ischemia and necrosis. They also contributed to data interpretation and revision of the manuscript.
- Andrés Redondo-Rodríguez organized and participated in the new optical mapping experiments we performed in isolated heart preparations from persistent AF animals. He also provided new animals and participated in the *in vivo* experiments for regional analysis of atrial electromechanical properties. He also contributed to data interpretation and revision of the manuscript.

We hope the reviewer can understand the multidisciplinary and translational approach of this work, which required the collaboration and participation of coauthors with different expertise.